# EMBEDDED-MODEL FLOWS: COMBINING THE INDUCTIVE BIASES OF MODEL-FREE DEEP LEARNING AND EXPLICIT PROBABILISTIC MODELING

**Gianluigi Silvestri**
OnePlanet Research Center
imec-the Netherlands
Wageningen, Netherlands
`gianluigi.silvestri@imec.nl`

**Emily Fertig, Dave Moore**
Google Research
San Francisco, CA, USA
`{emilyaf, davmre}@google.com`

**Luca Ambrogioni**
Department of Artificial Intelligence
Donders Institute for Brain, Cognition and Behaviour
Radboud University
Nijmegen, Netherlands
`l.ambrogioni@donders.ru.nl`

## ABSTRACT

Normalizing flows have shown great success as general-purpose density estimators. However, many real world applications require the use of domain-specific knowledge, which normalizing flows cannot readily incorporate. We propose *embedded-model flows* (EMF), which alternate general-purpose transformations with *structured* layers that embed domain-specific inductive biases. These layers are automatically constructed by converting user-specified differentiable probabilistic models into equivalent bijective transformations. We also introduce *gated structured layers*, which allow bypassing the parts of the models that fail to capture the statistics of the data. We demonstrate that EMFs can be used to induce desirable properties such as multimodality and continuity. Furthermore, we show that EMFs enable a high performance form of variational inference where the structure of the prior model is embedded in the variational architecture. In our experiments, we show that this approach outperforms a large number of alternative methods in common structured inference problems.

## 1 INTRODUCTION

Normalizing flows have emerged in recent years as a strong framework for high-dimensional density estimation, a core task in modern machine learning (Papamakarios et al., 2021; Kobyzev et al., 2020; Kingma & Dhariwal, 2018). As with other deep generative architectures such as GANs and VAEs, normalizing flows are commonly designed with the flexibility to model a very general class of distributions. This works well for large naturalistic datasets, such as those common in computer vision. However, general-purpose flows are less appealing when datasets are small or when we possess strong *a priori* knowledge regarding the relationships between variables.

On the other hand, differentiable (or 'deep') probabilistic programming is a powerful and general framework for explicit stochastic modeling that can express strong assumptions about the data-generating process (Tran et al., 2017; 2016; Bingham et al., 2019; Dillon et al., 2017; Piponi et al., 2020; Kucukelbir et al., 2017; Ambrogioni et al., 2021a). For example, the user may know that the data represents a process evolving over time or in space, or is subject to known physical laws, or that there is an unobserved common cause or other shared factor behind a set of observations. Such structured prior information can enable the user to form effective inferences using much less data than would otherwise be required. For example, in the field of astrophysics, differentiable probabilistic programs have been successfully used to study strong gravitational lensing (Chianese et al., 2020) and

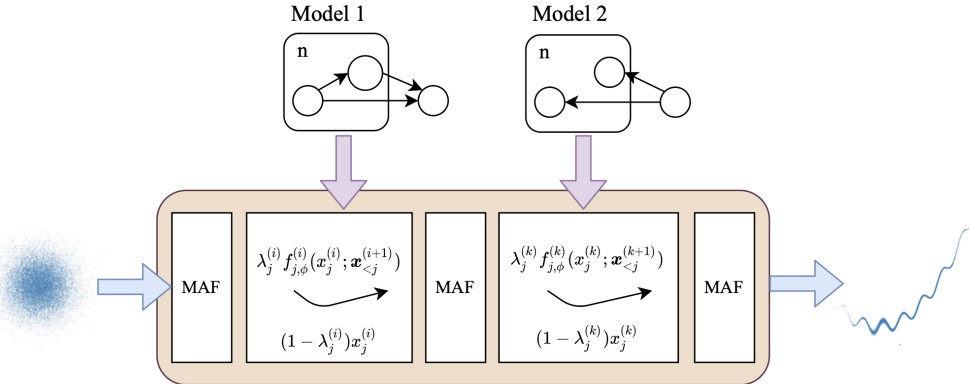

Figure 1: Didactic diagram of a hypothetical embedded-model architecture with model-free masked autoregressive flow (MAF) layers and two gated layers corresponding to two different probabilistic models.

dark matter substructures (Coogan et al., 2020; Varma et al., 2020). All of these applications entail the translation of sophisticated physical theories into highly parameterized and structured probabilistic programs. However, it remains the case that "all models are wrong" (Box, 1976), and it is rare that a modeler can be fully confident in the strong assumptions (e.g., Gaussianity, statistical independence) expressed by typical parametric probability models.

In this paper we bridge these two worlds, providing an automatic technique to convert differentiable probabilistic programs into equivalent normalizing flow layers with domain-specific inductive biases. We call these architectures *embedded-model flows* (EMF). EMFs can be use to integrate the structure of "wrong but useful" models within generic deep learning architectures capable of correcting the deviation of the model from the data. Furthermore, in the context of Bayesian inference, we show that the EMF architecture can be used to define a powerful class of approximate posterior distributions that embed the structure of the prior.

## 2 PRELIMINARIES

Normalizing flows define a random variable as an invertible differentiable transformation $\boldsymbol{y} = f_{\boldsymbol{\phi}}(\boldsymbol{x}^{(0)})$ of a base variable $\boldsymbol{x}^{(0)} \sim p_0(\boldsymbol{x}^{(0)})$. In these expressions, $\boldsymbol{\phi}$ are parameters that control the form of the transformation and consequently the distribution of $\boldsymbol{y}$. The transformed (log-)density can then be given explicitly using the change of variables formula from probability theory,

$$\log p(\boldsymbol{y}) = \log p_0(f_{\boldsymbol{\phi}}^{-1}(\boldsymbol{y})) - \log|\det J_{\boldsymbol{\phi}}(\boldsymbol{y})|, \tag{1}$$

where $J_{\boldsymbol{\phi}}(\boldsymbol{y})$ is the Jacobian matrix of $f_{\boldsymbol{\phi}}$. The base distribution $p_0$ is often taken to be standard normal ($\boldsymbol{x}^{(0)} \sim N(\boldsymbol{0}, \boldsymbol{I})$). In typical applications, the functional form of $f_{\boldsymbol{\phi}}$ is specified as the composition of a series of simpler invertible transformations:

$$\boldsymbol{y} = f_{\boldsymbol{\phi}}(\boldsymbol{x}^{(0)}) = \underbrace{f_{\boldsymbol{\phi}^{(0)}}^{(0)} \circ \cdots \circ f_{\boldsymbol{\phi}^{(m-1)}}^{(m-1)}}_{m \text{ times}}(\boldsymbol{x}^{(0)}) . \tag{2}$$

We denote the resulting sequence of transformed variables as $\boldsymbol{x}^{(0)}, \boldsymbol{x}^{(1)}, \ldots, \boldsymbol{x}^{(m-1)}, \boldsymbol{y}$. Each of these transformations (i.e. *flow layers*) are designed to have tractable formulas for inversion and log-Jacobian. The parameters $\boldsymbol{\phi}$ are usually trained by stochastic gradient descent to maximize the log-likelihood (given observed data) or evidence lower bound (in the variational inference setting).

### 2.1 PROBABILISTIC PROGRAMS AND DIRECTED GRAPHICAL MODELS

Probabilistic programming allows a joint probability distribution over a set of random variables to be specified intuitively as a program that includes random sampling steps (van de Meent et al., 2018).

Computations on this distribution, including the joint log-density function, structured inference algorithms, and the bijective transformations described in this paper, may then be derived automatically as program transformations. Recently frameworks have emerged for 'deep' probabilistic programming, which exploit the automatic differentiation and hardware acceleration provided by deep learning frameworks; examples include Pyro (Bingham et al., 2019), PyMC (Salvatier et al., 2016), Edward (Tran et al., 2017), and TensorFlow Probability (Dillon et al., 2017; Piponi et al., 2020). These allow for straightforward implementation of gradient-based inference and parameter learning, and enabling methods such as ours that integrate probabilistic programs with standard 'deep learning' components such as flows.

Probabilistic programs that contain only deterministic control flow may equivalently be viewed as directed graphical models, in which each random variable $\boldsymbol{x}_j$ is a graph node connected by incoming edges to some subset of the previous variables $\boldsymbol{x}_{<j}$. This leads to the following general expression for the joint density of a set of vector-valued variables $\boldsymbol{x}_{1:n} = \{\boldsymbol{x}_j\}_{j=1}^n$:

$$p(\boldsymbol{x}_{0:n}; \boldsymbol{\phi}) = p_0(\boldsymbol{x}_0; \boldsymbol{\theta}_0(\boldsymbol{\phi})) \prod_{j=1}^n p_j(\boldsymbol{x}_j | \boldsymbol{\theta}_j(\boldsymbol{x}_{<j}; \boldsymbol{\phi})) \tag{3}$$

Here $\boldsymbol{\theta}_j(\boldsymbol{x}_{<j}; \boldsymbol{\phi})$ denotes the 'link function' associated with each node that takes as input the model parameters $\boldsymbol{\phi}$ and the values of all parent variables (excepting $\boldsymbol{\theta}_0$, where there are no parents), and outputs the parameters of its respective density function.

As a practical matter, flow architectures generally assume layers of fixed size, so the experiments in this paper focus on programs of fixed structure and we use graphical-model notation for simplicity. However, the methods we describe do not require access to an explicit graph structure and could be straightforwardly applied to programs with stochastic control flow.

## 3 CONVERTING PROBABILISTIC PROGRAMS INTO NORMALIZING FLOWS

In this section, we will show that a large class of differentiable probabilistic programs can be converted into equivalent normalizing flow layers. More precisely, we will construct an invertible function $F_{\boldsymbol{\phi}}$ that maps spherical normal random inputs to the joint distribution of the probabilistic program:

$$\boldsymbol{x}^{(1)} = F_{\boldsymbol{\phi}}(\boldsymbol{x}^{(0)}) \sim p(\boldsymbol{x}^{(1)}; \boldsymbol{\phi}). \qquad (\boldsymbol{x}^{(0)} \sim \mathcal{N}(\boldsymbol{0}, \boldsymbol{I})) \tag{4}$$

This is a special case of the so called *Rosenblatt transform* (Rosenblatt, 1952). We interpret this transform as a flow layer and we extend it by including trainable gate variables that can be used to "disconnect" a set of variables from its parents so to correct model miss-specification. In our applications, the input to the bijective transformation may be arbitrary real-valued vectors, for example, the output of a previous flow layer, which likely will not be normally distributed. In such cases the transformed values will not be samples from the original probabilistic program, but we expect the transformation to capture some of its properties, such as multimodality and the dependence structure of its random variables.

### 3.1 INDIVIDUAL RANDOM VARIABLES AS BIJECTIVE TRANSFORMATIONS

We first consider the case of a univariate random variable $x^{(1)} \sim p_{\phi}(x^{(1)})$, which we can express as a continuous bijective transformation $f_{\phi}$ of a standard normal variable $x^{(0)}$ via a slight extension of inverse transform sampling. The probability integral transform theorem states that the cumulative distribution function (CDF) of $p_{\phi}$, $C_{\phi}(x) = \int_a^x p_{\phi}(y)\mathrm{d}y$, with $p_{\phi}$ defined over a generic interval $(a, b)$ and $a < x < b$, is uniformly distributed, $u = C_{\phi}(x) \sim \mathcal{U}(0, 1)$ (Angus, 1994). If the CDF is invertible, i.e., strictly increasing within its domain, it follows that we can sample $x = C_{\phi}^{-1}(u) \sim p_{\phi}$ given only a uniform random variable $u \sim \mathcal{U}(0, 1)$. We obtain $u$ through a second application of the probability integral transform theorem to the standard normal CDF $\Phi_{0,1}$. Our univariate 'flow' is therefore given by

$$x^{(1)} = f_{\phi}(x^{(0)}) = C_{\phi}^{-1}(\Phi_{0,1}(x^{(0)})) \qquad (x^{(0)} \sim \mathcal{N}(0, 1)) \tag{5}$$

Closed-form expressions for $f_{\phi}$ may be available in some cases (see below). In the general univariate setting where the inverse distribution function $C_{\phi}^{-1}(u)$ is not available in closed form, it may

be evaluated efficiently using numerical root-finding methods such as the method of bisection (see Appendix A.6), with derivatives obtained via the inverse function theorem and the implicit reparameterization trick (Figurnov et al., 2018). However, this is not needed for maximum likelihood training of the parameters $\phi$, since eq. (1) only involves $f_\phi^{-1}$ and its derivatives.

**Example: Gaussian variables**. The inverse CDF of a Gaussian variable $x^{(1)} \sim \mathcal{N}(\mu, \sigma^2)$ can be expressed as a scale-shift transformation of the inverse CDF of a standard normal variable: $\Phi_{\mu,\sigma}^{-1}(u) = \mu + \sigma \Phi_{0,1}^{-1}(u)$. We can obtain the forward flow transformation by composing this function with the CDF of a standard Gaussian: $f_{\mu,\sigma}(x^{(0)}) = \Phi_{\mu,\sigma}^{-1}(\Phi_{0,1}(x^{(0)})) = \mu + \sigma x^{(0)}$. This is the famous reparameterization formula used to train VAEs and other variational methods (Kingma & Welling, 2013). The inverse function is $f_{\mu,\sigma}^{-1}(x^{(1)}) = (x^{(1)} - \mu)/\sigma$ while the log-Jacobian is equal to $\log \sigma$.

It is not generally straightforward to evaluate or invert the CDF of a multivariate distribution. However, the most popular multivariate distributions, such as the multivariate Gaussian, can be expressed as transformations of univariate standard normal variables, and so may be treated using the methods of this section. More generally, the next section presents a simple approach to construct structured bijections (i.e., flows) from multivariate distributions expressed as probabilistic programs.

## 3.2 FROM DIFFERENTIABLE PROBABILISTIC PROGRAMS TO STRUCTURED LAYERS

A probabilistic program samples a set of variables sequentially, where (as in eq. (3)) the distribution of each variable is conditioned on the previously-sampled parent variables. Using the results of Section 3.1, we can represent this conditional sample as an invertible transformation of a standard normal input. Feeding the result back into the model defines a sequence of local transformations, which together form a multivariate flow layer (algorithm 1). More specifically, as with autoregressive flows, the forward transformation of the $j$-th input requires us to have computed the preceding outputs $\boldsymbol{x}_{<j}^{(i+1)}$. This may be expressed recursively as follows:

$$\boldsymbol{x}_{1:n}^{(i+1)} = f_{j,\phi}^{(i)}(\boldsymbol{x}_j^{(i)}; \boldsymbol{x}_{<j}^{(i+1)}), \tag{6}$$

where $f_{j,\phi}^{(i)}(\cdot; \boldsymbol{x}_{<j}^{(i+1)})$ denotes the invertible transformation associated to the conditional distribution of the $j$-th variable given the values of its parents. The recursive expression terminates since the parenting graph has an acyclic-directed structure.

In the reverse direction, the variables $\boldsymbol{x}_{1:n}^{(i+1)}$ are all given, so all transformations may be evaluated in parallel:

$$\boldsymbol{x}_{1:n}^{(i)} = f_{j,\phi}^{(i)-1}(\boldsymbol{x}_j^{(i+1)}; \boldsymbol{x}_{<j}^{(i+1)}) \tag{7}$$

This is a generalization of the inversion formula used in autoregressive flows (Kingma et al., 2016; Papamakarios et al., 2017). The Jacobian of the transformation is block triangular (up to a permutation of the variables) and, consequently, the log determinant is just the sum of the log determinants of the block diagonal entries:

$$\log\left|\det J_\phi^{-1}(\boldsymbol{x}_{1:n}^{(i+1)})\right| = \sum_j \log\left|\det J_{j,\phi}^{-1}(\boldsymbol{x}_j^{(i+1)}; \boldsymbol{x}_{<j}^{(i+1)})\right| \tag{8}$$

where $J_{j,\phi}^{-1}(\boldsymbol{x}_j^{(i+1)}; \boldsymbol{x}_{<j}^{(i+1)})$ is the Jacobian of the local inverse transformation $f_{j,\phi}^{(i)-1}(\boldsymbol{x}_j^{(i+1)}; \boldsymbol{x}_{<j}^{(i+1)})$.

We denote this multivariate invertible transformation as a *structured layer*. If the input $\boldsymbol{x}_{1:n}^{(i)}$ follows a standard Gaussian distribution, then the transformed variable follows the distribution determined by the input probabilistic program. This is not going to be the case when the layer is placed in the middle of a larger flow architecture. However, it will still preserve the structure of the probabilistic program with its potentially complex chain of conditional dependencies.

**Gated structured layers** Most user-specified models offer a simplified and imperfect description of the data. In order to allow the flow to skip the problematic parts of the model, we consider *gated* layers. In a gated layer, we take each local bijective transformation $g_{j,\phi}$ to be a convex combination of the original transformation $f_{j,\phi}$ and the identity mapping:

$$g_{j,\phi}^{(i)}(x_j^{(i)}; \boldsymbol{x}_{<j}^{(i+1)}, \lambda_j^{(i)}) = \lambda_j^{(i)} f_{j,\phi}^{(i)}(x_j^{(i)}; \boldsymbol{x}_{<j}^{(i+1)}) + (1 - \lambda_j^{(i)}) x_j^{(i)} \tag{9}$$

---

**Algorithm 1** Gated forward transformation: $\phi$: Model parameters; $\{\boldsymbol{x}_k^{(i)}\}_{k=1}^n$: Input variables sorted respecting parenting structure; $\{\lambda_k^{(i)}\}_{k=1}^n$: Gate parameters; $n$ Number of variables.

---

**procedure** GATEDFORWARDSAMPLER($\{\boldsymbol{x}_k^{(i)}\}_{k=1}^n$) State $\log|\det J| \leftarrow 0$
    **for** $j$ in Range($n$) **do**
        $\{p_l\}_{l=1}^{m_j} \leftarrow \boldsymbol{x}_j^{(i)}$.parents_idx              $\triangleright$ $\{p_l\}_{l=1}^{m_j}$: indices of parents of $j$-th variable.
                                            $\triangleright$ $m_j$: number of parents of $j$-th variable.

        **if** $\{p_l\}_{l=1}^{m_j}$ is not empty **then**
            $\tilde{\boldsymbol{x}}_j^{(i+1)} \leftarrow f_{j,\phi}^{(i)}(\boldsymbol{x}_j^{(i)}; \{\boldsymbol{x}_{p_l}^{(i+1)}\}_{l=1}^{m_j})$
            $\boldsymbol{x}_j^{(i+1)} \leftarrow \lambda_j^{(i)}\tilde{\boldsymbol{x}}_j^{(i+1)} + (1-\lambda_j^{(i)})\boldsymbol{x}_j^{(i)}$
            $\log|\det J| \leftarrow \log|\det J| + \log\left(\det|J_{j,\phi}(\{\boldsymbol{x}_k^{(i+1)}\}_{k=1}^n) + (1-\lambda_j^{(i)})I|\right)$
        **else**
            $\tilde{\boldsymbol{x}}_j^{(i+1)} \leftarrow f_{j,\phi}^{(i)}(\boldsymbol{x}_j^{(i)}; \boldsymbol{\phi}_j)$          $\triangleright$ $\boldsymbol{\phi}_j$: parameters of the $j$-th root distribution.
            $\boldsymbol{x}_j^{(i+1)} \leftarrow \lambda_j^{(i)}\tilde{\boldsymbol{x}}_j^{(i+1)} + (1-\lambda_j^{(i)})\boldsymbol{x}_j^{(i)}$
            $\log|\det J| \leftarrow \log|\det J| + \log\left(\det|J_{j,\phi_0^j}(\boldsymbol{x}_j^{(i+1)}) + (1-\lambda_j^{(i)})I|\right)$
    **return** $\{\boldsymbol{x}_k^{(i+1)}\}_{k=1}^n$, $\log|\det J|$

---

This gating technique is conceptually similar to the method used in highway networks (Srivastava et al., 2015), recurrent highway networks (Zilly et al., 2017) and, more recently, highway flows (Ambrogioni et al., 2021b). Here the gates $\lambda_j^{(i)} \in (0,1)$ are capable of decoupling each node from its parents, bypassing part of the original structure of the program. As before, the local inverses $g_\phi^{-1}(\boldsymbol{x}_j^{(i+1)}; \boldsymbol{x}_{<j}^{(i+1)}, \lambda_j^{(i)})$ can in general be computed by numeric root search, but the Gaussian case admits a closed form in both directions:

$$g_{\mu,\sigma}(x^{(i)}, \lambda) = (1 + \lambda(\sigma-1))x^{(i)} + \lambda\mu; \qquad g_{\mu,\sigma}^{-1}(x^{(i+1)}, \lambda) = \frac{x^{(i+1)} - \lambda\mu}{1 + \lambda(\sigma-1)} \qquad (10)$$

Analogues of eq. (6), eq. (7), and eq. (8) then define the gated joint transformation, its inverse and log Jacobian determinant, respectively. Note that the gating affects only the diagonal of the Jacobian, so the overall block-triangular structure is unchanged.

## 3.3 AUTOMATION

All the techniques described in this section can be fully automated in modern deep probabilistic programming frameworks such as TensorFlow Probability. A user-specified probabilistic program in the form given by Eq. 3 is recursively converted into a bijective transformation as shown in the pseudo-code in Fig. 1. In TensorFlow Probability, the *bijector* abstraction uses pre-implemented analytic inverse and Jacobian formulas and can be adapted to revert to root finding methods otherwise.

## 3.4 EMBEDDED-MODEL FLOWS

A EMF is a normalizing flow architecture that contains one or more (gated) structured layers. EMFs can be used to combine the inductive biases of explicit models with those of regular normalizing flow architectures. Several probabilistic programs can be embedded in a EMF both in parallel or sequentially. In the parallel case, a single flow layer is comprised by two or more (gated) structured layers acting on non-overlapping subsets of variables. The use of several programs embedded in a single architecture allows the model to select the most appropriate structure during training by changing the gating parameters and the downstream/upstream transformations. This allows the user to start with a "wrong but useful model" and then learn the deviation from the distribution of the collected data using flexible generic normalizing flow transformations. A visualization of a hypothetical EMF architecture is given in Fig. 1.

## 4    RELATED WORK

Embedded-model flows may be seen as a generalization of autoregressive flows (Kingma et al., 2016; Papamakarios et al., 2017) in the special case where the probabilistic program consists of a Gaussian autoregression parameterized by deep link networks, although generic autoregressive architectures often permute the variable order after each layer so that the overall architecture does not reflect any specific graphical structure. Graphical normalizing flows (Wehenkel & Louppe, 2021) generalize autoregressive structure to arbitrary DAG structures; however, this approach only constrains the conditional independence structure and does not embed the forward pass of a user-designed probabilistic program.

In the last few years, the use of tailored model-based normalizing flow architectures has gained substantial popularity. Perhaps the most successful example comes from physics, where gauge-equivariant normalizing flows are used to sample from lattice field theories in a way that guarantees preservation of the intricate symmetries of fundamental fields (Albergo et al., 2019; Kanwar et al., 2020; Albergo et al., 2021; Nicoli et al., 2021). Similarly, structured flows have been used to model complex molecules (Satorras et al., 2021), turbulence in fluids (Bezgin & Adams, 2021), classical dynamical systems (Rezende et al., 2019; Li et al., 2020) and multivariate timeseries (Rasul et al., 2020). These approaches are tailored to specific applications and their design requires substantial machine learning and domain expertise.

Normalizing flows have been applied to variational inference since their introduction by Rezende & Mohamed (2015). Recognizing that structured posteriors can be challenging for generic architectures, a number of recent approaches attempt to exploit the model structure. Structured conditional continuous normalizing flows (Weilbach et al., 2020) use minimally faithful inversion to constrain the posterior conditional independence structure, rather than embedding the forward pass of the prior model. Our architecture can be seen as a flexible extension of previous variational models that also embed the prior structure, such as stochastic structured variational inference (Hoffman & Blei, 2015), automatic structured variational inference (Ambrogioni et al., 2021a) and cascading flows (Ambrogioni et al., 2021b). It is also closely related to the non-centering transformations applied by Gorinova et al. (2020) to automatically reparameterize a probabilistic program, although that work focused on improving inference geometry and did not directly exploit the bijective aspect of these reparameterizations.

## 5    APPLICATIONS

Model-embedding allows users of all levels of expertise to insert domain knowledge into normalizing flow architectures. We will illustrate this with some examples, starting from simple uncoupled multi-modal models and moving on to more sophisticated structured models for time series. Finally, we will discuss the application of EMF to automatic structured variational inference, where we use EMF architectures that embed the prior distribution. Details about the datasets used, the models' hyperparameters and the training procedures can be found in Appendix A.1, A.3 and A.4 respectively. Additional experiments with an hierarchical structure can be found in appendix A.7.3. Number of parameters and sampling time for all the models are reported in appendix A.5. All the code used in the experiments is available at `https://github.com/gisilvs/EmbeddedModelFlows`.

### 5.1    MULTIMODALITY

Multimodal target densities are notoriously hard to learn for traditional normalizing flow architectures (Cornish et al., 2020). In contrast, it is straightforward to model multimodal distributions using explicit mixture models; the simplest example is perhaps a model where each variable follows an independent mixture of Gaussian distributions:

$$p_{\boldsymbol{\rho},\boldsymbol{\mu},\boldsymbol{\sigma}}(\boldsymbol{x}) = \prod_j \sum_k \rho_{j,k} \mathcal{N}(x_j; \mu_{jk}, \sigma_{jk}^2) \,. \tag{11}$$

The CDF of a mixture of distributions is given by the mixture of the individual CDFs. Therefore, the inverse transformation associated with a mixture of Gaussians model is

$$f^{-1}_{\boldsymbol{\rho},\boldsymbol{\mu},\boldsymbol{\sigma}}(x) = \Phi^{-1}_{0,1}\left(\sum_{j=1}^{n} \rho_j \Phi_{\mu,\sigma}(x)\right) \tag{12}$$

In itself, the probabilistic program in Eq. 12 can only model a very specific family of multivariate mixture distributions with Gaussian components organized in an axis-aligned grid. However, the model becomes much more flexible when used as a structured layer in an EMF architecture since the generic upstream layers can induce statistical coupling between the input variables, thereby loosening the independence structure of the original model. The resulting approach is conceptually similar to architectures based on monotonic transformations such as neural spline flows (Durkan et al., 2019a;b), which are arguably more efficient as they have an analytic inversion formula. However, this application shows that a specific inductive bias such as multimodality can be effectively incorporated into a flow architecture by users with only a basic understanding of probabilistic modeling.

We apply an EMF with an independent mixture of Gaussians to two 2D toy problems common in the flow literature, namely the "eight Gaussians" and the "checkerboard", and to the MNIST dataset. As baselines, we use a Masked Autoregressive Flow (MAF) model with two autoregressive layers and a standard Gaussian as a base density, a "large" version of MAF (MAF-L) with three autoregressive layers, which has much more flexibility compared to the other models in terms of trainable parameters, and a six steps Neural Splines Flows (NFS, with Rational Quadratic splines) with coupling layers as proposed in (Durkan et al., 2019b). We then combine the EMF with the MAF baseline by adding an EMF on top of the two autoregressive layers (EMF-T, for top) and in between the two layers, after the latent variables permutation (EMF-M, for middle). We also combine NSF with the structured layer, both with the EMF-T and EMF-M settings. Each mixture has 100 components, with trainable weights, means and standard deviations. The results are reported in table 1, while densities are shown in appendix A.7.1. The use of the multimodal structured layer with MAF always outperforms the autoregressive models. The improvement is not solely due to the increased parameterization as both methods outperform the more complex baseline MAF-L. NSF achieves superior results, as it is an architecture designed to model multimodality. The combination of the structured layer with NSF achieves the overall best performances. However, this comes at the cost of additional sampling time, caused by the numeric inversion of the mixture of Gaussians transformation.

Table 1: Results in negative log probability on one million test samples for the 2d toy problems. We report mean and standard error of the mean over five different runs. 8G stands for eight gaussians while CKB for checkerboard. The resulting negative log probability for MNIST is computed on the test set.

| | EMF-T | EMF-M | NSF-EMF-T | NSF-EMF-M | MAF | MAF-L | NSF |
|---|---|---|---|---|---|---|---|
| 8G | $2.881 \pm 0.000$ | $2.897 \pm 0.002$ | $2.837 \pm 0.003$ | $\mathbf{2.829 \pm 0.006}$ | $3.341 \pm 0.009$ | $3.032 \pm 0.018$ | $2.832 \pm 0.004$ |
| CKB | $3.579 \pm 0.000$ | $3.503 \pm 0.000$ | $\mathbf{3.480 \pm 0.001}$ | $3.494 \pm 0.009$ | $3.795 \pm 0.000$ | $3.614 \pm 0.006$ | $3.481 \pm 0.002$ |
| MNIST | $999.541 \pm 0.504$ | $1024.085 \pm 1.285$ | $\mathbf{958.960 \pm 0.695}$ | $961.078 \pm 0.316$ | $1310.943 \pm 0.676$ | $1291.870 \pm 0.951$ | $969.393 \pm 0.838$ |

## 5.2 TIMESERIES MODELING

Timeseries data is usually characterized by strong local correlations that can often be described with terms such as continuity and smoothness. In the discrete models considered here, these properties can be obtained by discretizing stochastic differential equations. One of the simplest timeseries model with continuous path is the Wiener process $\dot{x}(t) = w(t)$, with $w(t)$ being a white noise input with variance $\sigma^2$. By Euler–Maruyama discretization with $\Delta t = 1$, we obtain the following autoregressive model

$$x_{t+1} \sim \mathcal{N}\big(x_{t+1}; x_t, \sigma^2\big) \ , \tag{13}$$

which gives us the bijective function $f_{t+1,\sigma}(x_{t+1}; x_t) = x_t + \sigma x_{t+1}$. We use four stochastic differential equation models as time series datasets, discretized with the Euler-Maruyama method: the models are Brownian motion (BR), Ornstein-Uhlenbeck (OU) process, Lorenz system (LZ), and Van der Pol oscillator (VDP). We use an EMF with continuity structure. The baselines are MAF, MAF-L and NSF models like in the previous section, and an MAF in which the base distribution follows the continuity structure (B-MAF), as a discrete-time version of the model proposed in (Deng

et al., 2020). Note that no variables permutation is used for B-MAF. We then use GEMF-T and a combination of NSF with the structured layer (NSF-GEMF-T), with continuity structure. The results are reported in Table 2. The baselines combined with continuity structure greatly outperforms the other models (even the more complex MAF-L) for all of the datasets. Results obtained with the smoothness structure are reported in appendix A.7.4.

Table 2: Results in negative log probability on one million test samples for the time-series toy problems, and on a test set of 10000 datapoints. We report mean and standard error of the mean over five different runs.

|     | GEMF-T(c) | NSF-GEMF-T(c) | MAF | MAF-L | NSF | B-MAF |
|-----|-----------|---------------|-----|-------|-----|-------|
| BR  | $\mathbf{-26.414 \pm 0.0115}$ | $-26.221 \pm 0.0264$ | $-26.061 \pm 0.0103$ | $-26.121 \pm 0.0099$ | $-26.195 \pm 0.0565$ | $-25.905 \pm 0.0100$ |
| OU  | $\mathbf{24.086 \pm 0.0015}$ | $27.274 \pm 0.0858$ | $24.197 \pm 0.0028$ | $24.162 \pm 0.0029$ | $28.425 \pm 0.0499$ | $24.169 \pm 0.0034$ |
| LZ  | $-221.304 \pm 0.2253$ | $\mathbf{-224.981 \pm 0.2152}$ | $-194.240 \pm 0.3240$ | $-201.734 \pm 0.9729$ | $-203.270 \pm 0.4950$ | $-218.351 \pm 0.6960$ |
| VDP | $\mathbf{-562.081 \pm 0.0446}$ | $-558.939 \pm 0.1478$ | $-517.159 \pm 0.2567$ | $-558.032 \pm 0.2753$ | $-533.506 \pm 0.7265$ | $-557.420 \pm 0.1056$ |

## 5.3 Variational inference

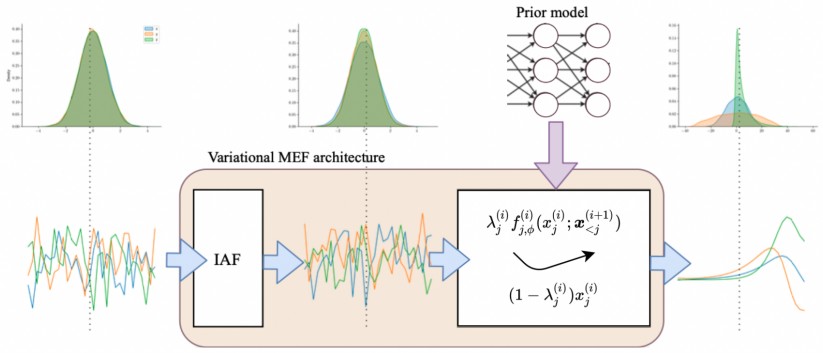

Figure 2: Diagram of a gated prior-embedding architecture with a sample of transformed variables for a Lorentz dynamics smoothing problem.

Normalizing flows are often used as surrogate posterior distributions for variational inference in Bayesian models, where the target posterior may contain dependencies arising both from the prior structure and from conditioning on observed data via collider (or "explaining away") phenomena. Generic flow architectures must learn the prior structure from scratch, which wastes parameters and computation and may lead the optimization into suboptimal local minima. Embedding the prior model into the flow itself as a gated structured layer avoids this difficulty and should allow the remaining layers to focus their capacity on modeling the posterior.

We deploy a gated EMF architecture with a two-layers of inverse autoregressive flow (IAF) and a final structured layer embedding the prior model. This is visualized in 2 for inference in a Lorentz dynamical system. In addition, we train plain IAF, automatic differentiation Mean Field (MF) and Multivariate Normal (MVN) surrogate posteriors from Kucukelbir et al. (2017) and ASVI from Ambrogioni et al. (2021a). We experiment on both time series and hierarchical models. The time series models are Brownian motion (BR), Lorenz system (LZ) and Van der Pol oscillator (VDP), with Bernoulli (classification) emissions. Furthermore, we either observe all the emissions (smoothing setting) or we omit a central time window (bridge setting). As hierarchical models we use the Eight Schools (ES) dataset and a Gaussian binary tree model with different depths and link functions, in which only the last node is observed (Ambrogioni et al., 2021b). The results are shown in Table 3. Results on additional experiments can be found in Appendix A.7.5. GEMF-T outperforms the baselines on all the problems (on-par with some baselines on the most trivial problems, see Appendix A.7.5), showing the benefits of embedding the prior program for highly structured problems, and the capability of such architecture to model collider dependencies.

Table 3: Results in negative ELBO. We report mean and standard error of the mean over ten different runs. S and B correspond respectively to smoothing and bridge, while c stands classification. As an example, BRB-c is the Brownian motion with the bridge-classification settings. For the binary tree experiments, Lin and Tanh correspond to the link function (Lin is linear link), and the following number is the depth of the tree.

|        | GEMF-T | MF | MVN | ASVI | IAF |
|--------|--------|----|-----|------|-----|
| BRS-c  | $\mathbf{15.882 \pm 1.405}$ | $23.764 \pm 1.262$ | $16.148 \pm 1.394$ | $15.894 \pm 1.403$ | $15.950 \pm 1.393$ |
| BRB-c  | $\mathbf{11.880 \pm 0.990}$ | $20.174 \pm 0.883$ | $12.098 \pm 0.985$ | $11.884 \pm 0.987$ | $11.947 \pm 0.985$ |
| LZS-c  | $\mathbf{8.317 \pm 1.033}$ | $60.458 \pm 0.511$ | $43.024 \pm 0.621$ | $26.296 \pm 2.008$ | $27.604 \pm 0.717$ |
| LZB-c  | $\mathbf{5.819 \pm 0.492}$ | $53.418 \pm 0.394$ | $36.063 \pm 0.449$ | $19.318 \pm 1.722$ | $20.778 \pm 0.542$ |
| VDPS-c | $\mathbf{68.385 \pm 2.599}$ | $162.336 \pm 1.425$ | $183.020 \pm 1.676$ | $70.884 \pm 2.247$ | $88.284 \pm 1.437$ |
| VDPB-c | $\mathbf{43.314 \pm 2.039}$ | $138.103 \pm 1.766$ | $157.736 \pm 2.146$ | $49.415 \pm 1.819$ | $63.914 \pm 1.502$ |
| ES     | $\mathbf{36.140 \pm 0.004}$ | $36.793 \pm 0.039$ | $36.494 \pm 0.014$ | $36.793 \pm 0.039$ | $36.169 \pm 0.007$ |
| Lin-8  | $\mathbf{2.596 \pm 0.213}$ | $108.869 \pm 0.467$ | $13.834 \pm 0.271$ | $26.232 \pm 0.340$ | $3.440 \pm 0.246$ |
| Tanh-8 | $\mathbf{1.626 \pm 0.159}$ | $144.668 \pm 0.349$ | $44.230 \pm 0.258$ | $14.241 \pm 0.650$ | $4.127 \pm 0.220$ |

## 6 DISCUSSION

We introduced EMF, a technique to combine domain-specific inductive biases with normalizing flow architectures. Our method automatically converts differentiable probabilistic programs into bijective transformations, allowing users to easily embed their knowledge into their models. We showed how, by choosing appropriate inductive biases, EMF can improve over generic normalizing flows on a range of different domains, with only a negligible increase in complexity, and we achieved high performance on a series of structured variational inference problems. EMF is a powerful generic tool which can find several applications both in probabilistic inference and more applied domains. **Limitations: I)** While we can map a very large family of probabilistic models to normalizing flow architectures, the approach is still limited by the topological constraints of bijective differentiable transformations (Cornish et al., 2020). This forces the support of the program to be homeomorphic to the range, making impossible to map models with non-connected support sets into normalizing flows. **II)** While the root finding methods used for inverting univariate densities are reliable and relatively fast, they do have higher computational demands than closed form inversion formulas. **III)** While it is possible to model discrete variables using Eq. 6, this does not result in an invertible and differentiable transformation that can be trained using the standard normalizing flow approach. **IV)** The probabilistic program must be executed during sampling. This can result in significant sampling slow-downs when using complex autoregressive programs. Note that, except in the case of variational inference, this does not lead to training slow-downs as the inverse transform is always trivially parallelizable. **Future work:** Embedded-model flows are a generic tool that can find application in several domains in which there is structure in the data. In theory, any number of structured layers can be added to EMF architectures, and combining different structures in the same model may help model complex dependencies in the data. Further empirical work is needed to asses the capacity of trained EMFs to select the correct (or the least wrong) model. While we only considered the use of structured layers in normalizing flows, they may also be used in any kind of deep net in order to embed domain knowledge in tasks such as classification, regression and reinforcement learning.

## ACKNOWLEDGEMENTS

OnePlanet Research Center aknowledges the support of the Province of Gelderland.

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

# A APPENDIX

## A.1 DATASETS

Here we provide a detailed description of the datasets used in the experiments.

### A.1.1 2D TOY PROBLEMS

The implementation of the 2D toy datasets is taken from the publicly available implementation of FFJORD (Grathwohl et al., 2019).

### A.1.2 MNIST

We use as training set 50000 data points, while the remaining 10000 are used as validation set. The images are dequantized following the same procedure as in (Papamakarios et al., 2017).

### A.1.3 STOCHASTIC DIFFERENTIAL EQUATIONS

For the generative time series experiments we generate data using four stochastic differential equations (SDEs) models, discretized with the Euler-Maruyama method. The SDEs are: Geometric Brownian motion, Ornstein-Uhlenbeck process, Lorenz system and Van der Pol oscillator. In the following, $W_t$ corresponds to a Wiener process, and the series length $T = 30$ with time step $s = 1$ unless specified.

**Geometric Brownian motion**: the dynamics evolve as:

$$\dot{x}_t = \mu x_t dt + \sigma x_t dW_t$$

where $\mu$ and $\sigma$ are constants. In practice, we generate sequences as a simple Brownian motion, and then take the exponential of the whole sequence. The Brownian motion sequence is generated as:

$$x_0 \sim \mathcal{N}(\mu, \sigma)$$
$$x_t \sim \mathcal{N}(x_{t-1}, \sigma), \forall t \in [1, \ldots, T-1]$$

where the mean $\mu = 0$ and the standard deviation $\sigma = 0.1$.

**Ornstein-Uhlenbeck process**: This process has the following dynamics:

$$\dot{x}_t = -\theta x_t dt + \sigma dW_t$$

with $\theta > 0$ and $\sigma > 0$. The sequences are generated as:

$$x_0 \sim \mathcal{N}(\mu, \sigma_0)$$
$$x_t \sim \mathcal{N}(\theta x_{t-1}, \sigma), \forall t \in [1, \ldots, T-1]$$

with $\mu_0 = 0$, $\sigma_0 = 5$, $\sigma = 0.5$ and $\theta = 0.8$.

**Lorenz system**: the Lorenz dynamics evolve as follows:

$$\dot{x}_t = \phi(y_t - x_t)$$
$$\dot{y}_t = x_t(\rho - z_t) - y_t$$
$$\dot{z}_t = x_t y_t - \beta z_t$$

where $\phi = 10$, $\rho = 28$ and $\beta = \frac{8}{3}$. In the discrete case we have:

$$x_0, y_0, z_0 \sim \mathcal{N}(0, 1)$$
$$\forall t \in [1, \ldots, T-1]$$
$$x_t \sim \mathcal{N}(x_{t-1} + s(\phi(y_{t-1} - x_{t-1})), \sqrt{s} * \sigma)$$
$$y_t \sim \mathcal{N}(y_{t-1} + s(x_{t-1}(\rho - z_{t-1}) - y_{t-1}), \sqrt{s} * \sigma)$$
$$z_t \sim \mathcal{N}(z_{t-1} + s(x_{t-1}y_{t-1} - \beta z_{t-1}), \sqrt{s} * \sigma)$$

with standard deviation $\sigma = 0.1$ and step size $s = 0.02$.

**Van der Pol oscillator**: The system evolves as:

$$\dot{x} = y + W_t$$
$$\dot{y} = \mu(1 - x^2)y - x + W_t$$

In the discrete case, with $T = 120$ and $s = 0.05$, we have:

$$x_0, y_0 \sim \mathcal{N}(0, 1)$$
$$\forall t \in [1, \ldots, T-1]$$
$$x_t \sim \mathcal{N}(x_{t-1} + y_{t-1} * s, \sqrt{s} * \sigma)$$
$$y_t \sim \mathcal{N}(y_{t-1} + s\mu(1 - x_{t-1}^2)y_{t-1}, \sqrt{s} * \sigma)$$

Here we use $\mu = 1$ and $\sigma = 0.1$

### A.1.4 SMOOTHING AND BRIDGE DATA

The data for the variational inference experiments is generated with either a Brownian motion, a Lorenz system or the Van der Pol oscillator. The first follows the same dynamics of the geometric Brownia motion described above, but without taking the exponential, while the others are the same as the Lorenz system and Van der Pol oscillator described above. The true time series remains unobserved, and we only have access to noisy emissions. In the regression case, the emissions follow a Gaussian distribution:

$$e_t \sim \mathcal{N}(x_t, \sigma_e)$$

where $\sigma_e$ is the emission noise, and is $\sigma_e = 0.15$ for the Brownian motion, $\sigma_e = 1$ for the Lorenz system, and $\sigma_e = 0.5$ for the Van der Pol oscillator. In the classification case, the emissions follow a Bernoulli distribution:

$$e_t \sim Bernoulli(kx_t)$$

with $k$ a gain parameter, $k = 5$ in the Brownian Motion, $k = 2$ in the Lorenz system and $k = 1$ for the Van der Pol oscillator. In the smoothing problem, we observe emission for all the time steps, while in the bridge problem we observe emissions only in the first and last 10 time points (first and last 40 for the Van der Pol oscillator). For the Lorenz system amd Van der Pol oscillator, the emissions are observed only for the $x$ variables in both smoothing and bridge.

### A.1.5 HIERARCHICAL MODELS

**IRIS dataset**: We use the IRIS dataset from the UCI Machine learning Repository. The dataset is composed by three classes of iris plant, each described by four attributes: sepal length, sepal width, petal length and petal width. There are 50 datapoints for each class. In the experiments, we use samples of $n = 10$ datapoints.

**Digits dataset**: We use the Digits dataset from the UCI Machine learning Repository. The dataset is composed by 10 classes $8 \times 8$ images of handwritten digits, from 0 to 9. In the experiments, we use samples of $n = 20$ datapoints. The images are dequantized following the same procedure as in (Papamakarios et al., 2017).

**Eight Schools**: The Eight Schools (ES) model considers the effectiveness of coaching programs on a standardized test score conducted in parallel at eight schools. It is specified as follows:

$$\mu \sim \mathcal{N}(0, 100)$$
$$\log \tau \sim \log \mathcal{N}(5, 1)$$
$$\theta_i \sim \mathcal{N}(\mu, \tau^2)$$
$$y_i \sim \mathcal{N}(\theta_i, \sigma_i^2)$$

where $\mu$ represents the prior average treatment effect, $\tau$ controls how much variance there is between schools, $i = 1, \ldots, 8$ is the school index, and $y_i$ and $\sigma_i$ are observed.

**Gaussian Binary tree**: The Gaussian Binary tree is a reverse tree of $D$ layers, in which the variables at a given layer $d$, $x_j^d$, are sampled from a Gaussian distribution where the mean is function of two parent variables $\pi_{j,1}^{d-1}, \pi_{j,2}^{d-1}$ at the previous layer:

$$x_j^d \sim \mathcal{N}(link(\pi_{j,1}^{d-1}, \pi_{j,2}^{d-1}), \sigma^2)$$

All the variables at the 0-th layers are sampled from a Gaussian with mean 0, and variance $\sigma^2 = 1$. The last node of the tree is observed, and the inference problem consists in computing the posterior of the nodes at the previous layers. We use a linear coupling function $link(x, y) = x - y$ and a nonlinear one $link(x, y) = \tanh(x) - \tanh(y)$, with trees of depth $D = 4$ and $D = 8$.

## A.2 STRUCTURES INDUCED WITH EMF

In this section we describe in detail the structures embedded with EMF and the bijective transformations they correspond to. In the following, we denote as $\bar{x}$ the output vector from the normalizing flow, and $\bar{x}'$ the vector after the transformation induced by EMF. When the probabilistic program embedded by EMF has only Gaussian distributions $\mathcal{N}(\mu, \sigma^2)$, the bijective transformation is in the form:

$$f(x) = \mu + \sigma x$$
$$f^{-1}(x) = \frac{(x - \mu)}{\sigma}$$

while in the gated case, with $\lambda \in (0, 1)$ the transformation becomes:

$$g(x, \lambda) = \lambda\mu + x(\lambda(\sigma - 1) + 1)$$
$$g^{-1}(x, \lambda) = \frac{x - \lambda\mu}{1 + \lambda(\sigma - 1))}$$

In both time series and Gaussian hierarchical generative models, the embedded structures have only Gaussian distributions, so all the gated and non-gated bijective transformation will have the form described above. What changes in the structure is the way we define the means and variances of such structures.

**Continuity**: for the first variable of the sequence, we have $\mu = 0$, $\sigma = \sigma_s$, while for the remaining $T - 1$ variables $x_t \forall t \in [1, \ldots, T - 1]$, we have $\mu = x'_{t-1}$ and $\sigma = \sigma_s$. $\sigma_s$ is a trainable variable transformed with the softplus function, shared among all the latent variables.

**Smoothness**: the first two variables of the sequence have $\mu = 0$ and $\sigma = 1$. The remaining variables have $\mu = 2x'_t - x'_{t-1}$ and $\sigma = \sigma_s$, with $\sigma_s$ a trainable variable like the case above.

**Gaussian hierarchical model**: for the first variable of the sequence, corresponding to the mean of the remaining variables, we have $\mu = 0$ and $\sigma = 1$, while for the remaining variables, we set $\mu = x'_0$ and $\sigma = 1$.

## A.3 FLOW MODELS

In all the experiments, we use MAF for generative models and IAF for variational inference, both with two autoregressive layers (three for MAF-L) and a standard Gaussian as a base density. Each autoregressive layer has two hidden layers with 512 units each. We use ReLU nonlinearity for all the models but for IAF, MAF and MAF-L without EMF or GEMF, for which we empirically found Tanh activation to be more stable. The features permutation always happen between the two autoregressive layers. In the case of EMF-M and GEMF-M, the permutation happens before the structured layer. We found empirically that training the gates for GEMF is slow as the variables lie on an unbounded space before the Sigmoid activation. A possible solution is scaling the variables (by 100 in our experiments) before applying Sigmoid, which speeds-up the training for the gates, leading to better results.

## A.4 TRAINING DETAILS

In this section we describe the training procedure for the experiments. We train all the models with Adam optimizer (Kingma & Ba, 2014).

**2D toy experiments**: the models are trained for 500 thousand iterations, with learning rate 1e-4 and cosine annealing. The mixtures of Gaussians are initialized with means evenly spaced between -4 and 4 for EMF-T and between -10 and 10 for EMF-M, while the standard deviations are initialized to 1 For EMF-T and 3 for EMF-M.

**MNIST**: the models are trained with early stopping until there is no improvement in the validation loss for 100 epochs, with learning rate 1e-4. The mixtures of Gaussians are initialized with means evenly spaced between -15 and 15 for EMF-T and between -20 and 20 for EMF-M, while the standard deviations are initialized to 3 For EMF-T and 1 for EMF-M.

**Generative Hierarchical**: we fit the models for 100 thousand iterations for the IRIS dataset and for 400 thousand iterations for the Digits dataset, with learning rate 1e-4 and cosine annealing. The annealing schedule is over 500 thousands iterations.

**Generative SDEs**: the models are trained for 50 thousand iterations on BR and OU, while for 400 thousand iterations for LZ and VDP, with learning rate 1e-4 and cosine annealing. The annealing schedule is over 500 thousand iterations.

**Variational inference**: we fit all the surrogate posteriors for 100000 iterations with full-batch gradient descent, using a 50-samples Monte Carlo estimate of the ELBO, and learning rate 1e-3 for all the methods but IAF, EMF-T and GEMF-T, for which we use 5e-5. For GEMF, the gates are initialized to be close to the prior, whit a value of 0.999 after Sigmoid activation.

### A.5  NUMBER OF PARAMETERS AND SAMPLE TIME

We report the number of trainable parameters and sample time in seconds for each model used in the generative experiments (tables 4, 5, 6, 7, 8, 9). Note that the samples were performed on a GPU Nvidia Quadro RTX 6000 . We do not report parameters and inference time for the Variational Inference experiments, as the number of parameter changes only with gates, in which there is one additional parameter per latent variable, and the inference time difference between models with and without EMF layer is similar to the sampling time in the generative case.

Table 4: Number of trainable parameters per model for the generative time series experiments.

|        | GEMF-T(c) | GEMF-T(s) | NSF-GEMF-T(c) | MAF     | MAF-L   | B-MAF   | NSF     |
|--------|-----------|-----------|---------------|---------|---------|---------|---------|
| BR/OU  | 618647    | 618647    | 291589        | 618616  | 927924  | 618617  | 291558  |
| LZ     | 803209    | 803209    | 861651        | 803176  | 1204764 | 803179  | 861618  |
| VDP    | 1264698   | 1264698   | 2286890       | 1264576 | 1896864 | 1264578 | 2286768 |

Table 5: Number of trainable parameters per model for the generative hierarchical experiments.

|        | GEMF-T  | GEMF-M  | MAF     | MAF-L   |
|--------|---------|---------|---------|---------|
| IRIS   | 649386  | 649386  | 649376  | 974064  |
| DIGITS | 4463636 | 4463636 | 4463616 | 6695424 |

Table 6: Number of trainable parameters per model for the 2D toy and MNIST experiments.

|                          | EMF-T   | EMF-M   | NSF-EMF-T | NSF-EMF-M | MAF     | MAF-L   | NSF     |
|--------------------------|---------|---------|-----------|-----------|---------|---------|---------|
| 8 Gaussians/Checkerboard | 533088  | 533088  | 26130     | 26130     | 532488  | 798732  | 25530   |
| MNIST                    | 3173120 | 3173120 | 7690512   | 7690512   | 2937920 | 4406880 | 7455312 |

Table 7: Sample time in seconds for the generative time series experiments. We report mean and standard deviation over ten samples with batch 100.

|  | GEMF-T(c) | GEMF-T(s) | NSF-GEMF-T(c) | MAF | MAF-L | B-MAF | NSF |
|---|---|---|---|---|---|---|---|
| BR | $0.658 \pm 0.048$ | $0.757 \pm 0.012$ | $0.310 \pm 0.007$ | $0.473 \pm 0.013$ | $0.704 \pm 0.053$ | $0.987 \pm 0.049$ | $0.154 \pm 0.010$ |
| OU | $0.655 \pm 0.038$ | $0.807 \pm 0.072$ | $0.338 \pm 0.040$ | $0.481 \pm 0.034$ | $0.664 \pm 0.041$ | $0.977 \pm 0.033$ | $0.184 \pm 0.006$ |
| LZ | $1.511 \pm 0.097$ | $1.636 \pm 0.077$ | $0.536 \pm 0.008$ | $1.283 \pm 0.039$ | $1.912 \pm 0.083$ | $1.834 \pm 0.067$ | $0.348 \pm 0.014$ |
| VDP | $4.419 \pm 0.170$ | $4.749 \pm 0.147$ | $1.063 \pm 0.071$ | $3.560 \pm 0.118$ | $5.337 \pm 0.234$ | $5.635 \pm 0.186$ | $0.378 \pm 0.011$ |

Table 8: Sample time in seconds for the generative hierarchical experiments. We report mean and standard deviation over ten samples with batch 100.

|  | GEMF-T | GEMF-M | MAF | MAF-L |
|---|---|---|---|---|
| IRIS | $0.667 \pm 0.046$ | $0.741 \pm 0.037$ | $0.595 \pm 0.024$ | $0.794 \pm 0.016$ |
| DIGITS | $18.040 \pm 0.335$ | $17.921 \pm 0.266$ | $17.395 \pm 0.232$ | $26.211 \pm 0.314$ |

Table 9: Sample time in seconds for the 2D toy and MNIST experiments. We report mean and standard deviation over ten samples with batch 100. We use batch 10 for MNIST as the numeric inversion needs a bigger amount of GPU memory.

|  | EMF-T | EMF-M | NSF-EMF-T | NSF-EMF-M | MAF | MAF-L | NSF |
|---|---|---|---|---|---|---|---|
| 8 Gaussians | $1.437 \pm 0.216$ | $1.409 \pm 0.236$ | $1.584 \pm 0.648$ | $1.526 \pm 0.241$ | $0.036 \pm 0.004$ | $0.050 \pm 0.005$ | $0.078 \pm 0.006$ |
| Checkerboard | $1.406 \pm 0.179$ | $1.379 \pm 0.135$ | $1.516 \pm 0.152$ | $1.546 \pm 0.178$ | $0.036 \pm 0.004$ | $0.049 \pm 0.004$ | $0.075 \pm 0.006$ |
| MNIST | $13.809 \pm 1.009$ | $13.991 \pm 0.844$ | $2.745 \pm 0.857$ | $3.791 \pm 0.518$ | $10.679 \pm 0.258$ | $16.272 \pm 0.495$ | $0.094 \pm 0.009$ |

### A.6 NUMERICAL ROOT FINDING METHODS

There are cases in which a closed form inversion for the transformation induced by EMF is not available, such as for the Mixture of Gaussian case. In those cases, we can still compute the inverse function and log-Jacobian determinants by using numerical root finding methods. In this work, we use either the secant method or the Chandrupatla's method (Chandrupatla, 1997; Scherer, 2010), as the implementation is available in the probabilistic programming framework Tensorflow Probability.

#### A.6.1 SECANT METHOD

The secant method is a root-finding algorithm that uses a succession of roots of secant lines to better approximate a root of a function. The secant method can be thought of as a finite-difference approximation of Newton's method. The secant method starts with two initial values $x_0$ and $x_1$ which should be chosen to lie close to the root, and then uses the following recurrent relation:

$$x_n = x_{n-1} - f(x_{n-1}) \frac{x_{n-1} - x_{n-2}}{f(x_{n-1}) - f(x_{n-2})}$$
$$= \frac{x_{n-2} f(x_{n-1}) - x_{n-1} f(x_{n-2})}{f(x_{n-1}) - f(x_{n-2})}$$

#### A.6.2 CHANDRUPATLA'S METHOD

This root-finding algorithm is guaranteed to converge if a root lies within the given bounds. At each step, it performs either bisection or inverse quadratic interpolation. The specific procedure can be found in (Chandrupatla, 1997; Scherer, 2010).

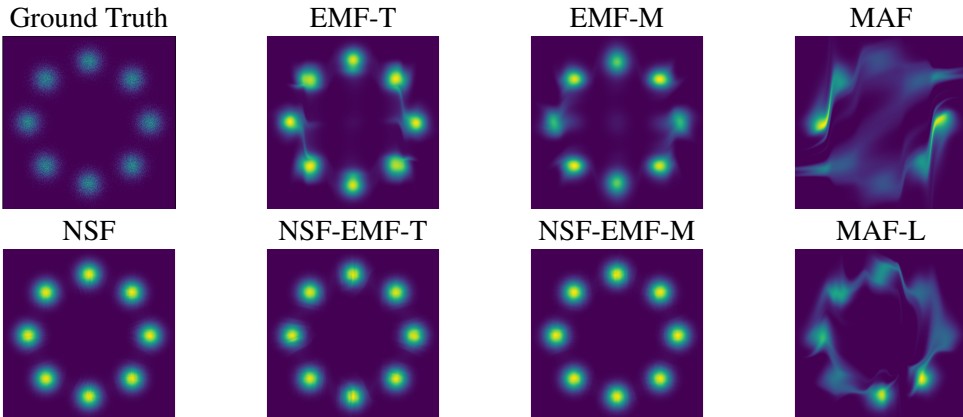

Figure 3: Comparison of densities learned by different models on the 8 Gaussians dataset.

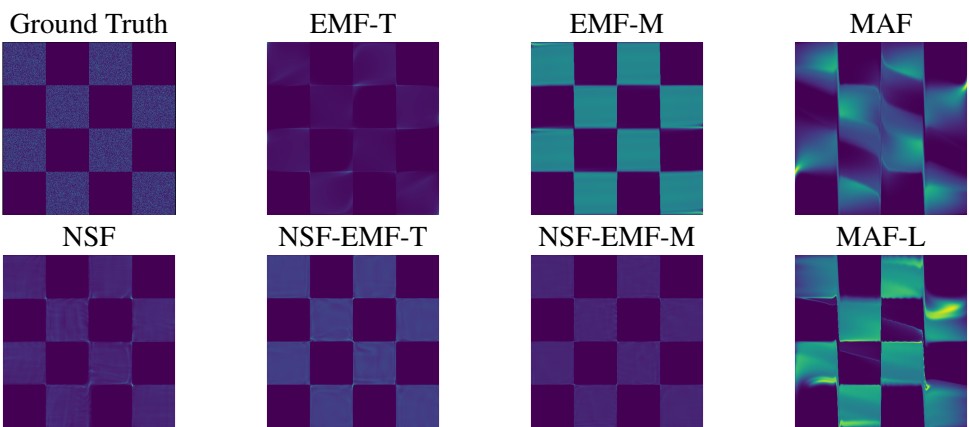

Figure 4: Comparison of densities learned by different models on the Checkerboard dataset.

## A.7 ADDITIONAL RESULTS

### A.7.1 DENSITY PLOTS FOR 2D TOY EXPERIMENTS

We include density plots for the 2D toy experiments in figures 3 and 4.

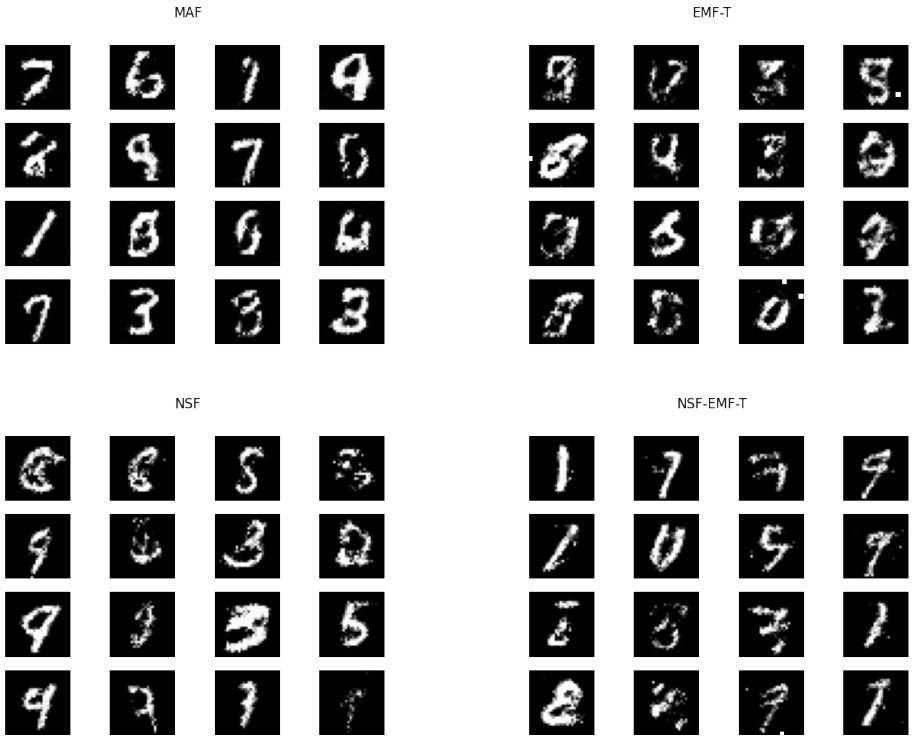

Figure 5: Sampled MNIST digits for different models.

### A.7.2 MNIST SAMPLES

We include samples for models trained on MNIST in figure 5.

### A.7.3    HIERARCHICAL MODELS

Hierarchical models are central to both Bayesian and frequentist statistics and can be used to model data with nested sub-populations. For example, a Gaussian hierarchical model with $m$ populations each with $n$ datapoints can be expressed as

$$p(\boldsymbol{m}, \boldsymbol{x}) = \prod_{k=1}^{m} \left( \mathcal{N}(m_k; 0, \sigma^2) \prod_{j=1}^{n-1} \mathcal{N}(x_{jk}; m_k, \nu^2) \right) \tag{14}$$

where $m_k$ denotes the empirical population mean. For the leaf variables, this results in the following structured layer: $f_{jk,\sigma}(x_{jk}, \eta_j) = \sigma\eta_k + \nu x_{jk}$ , where $x_{jk}$ is the latent variable of the leaf node $x_{jk}$ while $\eta_k$ is the latent of its root node. This extremely simple layer can be used to couple variables that are known to be hierarchically related and can be straightforwardly generalized to deeper and more complex hierarchical couplings. Again, the downstream and upstream layers in the EMF architecture can model additional couplings and non-Gaussianity while keeping the hierarchical structure. To test the effectiveness of EMF with Gaussian hierarchical structure, we train the flows on a hierarchical generative modeling problem where each network outputs a whole dataset subdivided into several classes. Specifically, we use two classification datasets, the IRIS dataset, in which three flower classes are described by four numerical features, and the Digits dataset, composed of $8 \times 8$ images of handwritten digits from 0 to 9. We modify the datasets by using vectors $\bar{x} = [\bar{\mu}, \bar{d}_1, \dots, \bar{d}_n]$, where the $\bar{d}_i, i \in [1, \dots, n]$ are datapoints randomly sampled from the same class, and $\bar{\mu}$ is the feature-wise mean of such datapoints. We then train the flows with vectors $\bar{x}' = [\bar{\mu}, \bar{d}_1, \dots, \bar{d}_{n-1}]$, as the last datapoint $\bar{d}_n$ can be inferred from the mean and the other $n-1$ datapoints. We use the same MAF and MAF-L baselines as for the 2d toy problems, and the gated version of EMF-T and EMF-M (GEMF-T, GEMF-M). The results are reported in table 10. The EMF architectures perform better than their MAF equivalents but worse than the much more highly parameterized MAF-L architectures. This is not surprising given the extreme simplicity of the hierarchical layer.

Table 10: Results in negative log probability on one hundred thousand test samples for the IRIS dataset and over ten thousand test samples for the Digits dataset. We report mean and standard error of the mean over five different runs.

|  | GEMF-T | GEMF-M | MAF | MAF-L |
|---|---|---|---|---|
| IRIS | $-9.165 \pm 0.046$ | $-8.095 \pm 0.054$ | $-7.213 \pm 0.066$ | $\boldsymbol{-11.446 \pm 0.3883}$ |
| DIGITS | $1260.849 \pm 1.0764$ | $1310.619 \pm 0.7419$ | $1307.927 \pm 0.5284$ | $\boldsymbol{1181.283 \pm 0.8151}$ |

### A.7.4    SMOOTHING STRUCTURE FOR TIME SERIES

We can induce smoothness using a layer obtained by discretizing the second-order equation $\ddot{x}(t) = -a\dot{x}(t) + w(t)$, which models a physical Brownian motion. We show the results obtained in table 11.

Table 11: Results in negative log probability on one million test samples for the time-series toy problems, and on a test set of n datapoints. We report mean and standard error of the mean over five different runs.

|  | GEMF-T(s) |
|---|---|
| BR | $-26.125 \pm 0.0015$ |
| OU | $24.131 \pm 0.0030$ |
| LZ | $-189.647 \pm 0.3899$ |
| VDP | $-518.464 \pm 2.5147$ |

### A.7.5    VI EXPERIMENTS

As additional variational inference experiments, we show the results on the time series with Gaussian (regression) emissions and a shallow version of the Gaussian binary tree model with depth 4. The

results are shown in tables 12. We also compute the forward KL divergence for each model. Such results are reported in table 13.

The EMF bijective transformation can be also used in combination with models which do not use normalizing flows. We combine EMF with the mean field and multivariate normal approximations from (Kucukelbir et al., 2017), with results in table 14. We also report results of the non gated version of EMF.

Table 12: Results in negative ELBO. We report mean and standard error of the mean over ten different runs

|  | GEMF-T | MF | MVN | ASVI | IAF |
|---|---|---|---|---|---|
| BRS-r | $-3.341 \pm 0.977$ | $0.147 \pm 1.002$ | $-3.126 \pm 0.977$ | $\mathbf{-3.354 \pm 0.974}$ | $-3.304 \pm 0.974$ |
| BRB-r | $-3.122 \pm 0.967$ | $1.932 \pm 0.951$ | $-2.866 \pm 0.967$ | $\mathbf{-3.142 \pm 0.966}$ | $-3.086 \pm 0.961$ |
| LZS-r | $\mathbf{46.981 \pm 0.476}$ | $1492.572 \pm 255.358$ | $1485.622 \pm 251.296$ | $1257.942 \pm 330.467$ | $1337.364 \pm 301.002$ |
| LZB-r | $\mathbf{110.500 \pm 40.958}$ | $756.338 \pm 146.012$ | $748.298 \pm 146.063$ | $490.696 \pm 166.080$ | $598.402 \pm 153.412$ |
| VDPS-r | $\mathbf{93.520 \pm 2.729}$ | $174.445 \pm 2.846$ | $194.592 \pm 2.181$ | $93.696 \pm 3.283$ | $117.028 \pm 2.666$ |
| VDPB-r | $\mathbf{68.030 \pm 1.315}$ | $150.184 \pm 1.445$ | $171.263 \pm 1.359$ | $72.401 \pm 3.076$ | $92.739 \pm 1.249$ |
| Lin-4 | $\mathbf{1.513 \pm 0.576}$ | $6.136 \pm 0.573$ | $1.542 \pm 0.571$ | $3.268 \pm 0.561$ | $1.526 \pm 0.581$ |
| Tanh-4 | $\mathbf{-0.035 \pm 0.119}$ | $6.640 \pm 0.157$ | $0.075 \pm 0.124$ | $2.866 \pm 0.113$ | $-0.026 \pm 0.118$ |

Table 13: Results in forward KL divergence. We report mean and standard error of the mean over ten different runs.

|  | GEMF-T | MF | MVN | ASVI | IAF |
|---|---|---|---|---|---|
| BRS-r | $37.362 \pm 1.049$ | $32.332 \pm 2.051$ | $\mathbf{37.608 \pm 1.122}$ | $37.346 \pm 1.091$ | $37.319 \pm 1.101$ |
| BRS-c | $28.539 \pm 0.920$ | $-32.940 \pm 9.829$ | $28.174 \pm 0.958$ | $\mathbf{28.571 \pm 0.913}$ | $28.458 \pm 0.896$ |
| BRB-r | $32.403 \pm 0.762$ | $11.608 \pm 6.311$ | $31.800 \pm 0.921$ | $\mathbf{32.449 \pm 0.748}$ | $32.267 \pm 0.769$ |
| BRB-c | $\mathbf{26.928 \pm 1.940}$ | $-76.227 \pm 48.668$ | $26.582 \pm 2.004$ | $26.825 \pm 1.946$ | $26.816 \pm 1.906$ |
| LZS-r | $\mathbf{245.458 \pm 2.404}$ | $-1.4e^{+08} \pm 2.3e^{+07}$ | $-7.6e^{+06} \pm 1.8e^{+06}$ | $-3.6e^{+04} \pm 1.3e^{+04}$ | $-1.0e^{+08} \pm 6.4e^{+07}$ |
| LZS-c | $\mathbf{245.337 \pm 1.577}$ | $-1.7e^{+08} \pm 1.9e^{+07}$ | $-1.1e^{+07} \pm 1.4e^{+06}$ | $-3.e^{+04} \pm 7.9e^{+03}$ | $-1.2e^{+06} \pm 7.1e^{+05}$ |
| LZB-r | $\mathbf{148.488 \pm 51.783}$ | $-1.2e^{+08} \pm 2.9e^{+07}$ | $-6.9e^{+06} \pm 1.9e^{+06}$ | $-4.2e^{+04} \pm 2.e^{+04}$ | $-2.1e^{+06} \pm 1.1e^{+06}$ |
| LZB-c | $\mathbf{249.709 \pm 1.348}$ | $-1.8e^{+08} \pm 2.1e^{+07}$ | $-1.1e^{+07} \pm 1.5e^{+06}$ | $-3.5e^{+04} \pm 6.9e^{+03}$ | $-7.7e^{+06} \pm 3.3e^{+06}$ |
| VDPS-r | $\mathbf{577.904 \pm 4.153}$ | $-4697.812 \pm 424.368$ | $390.420 \pm 28.331$ | $569.293 \pm 3.782$ | $-42.726 \pm 104.995$ |
| VDPS-c | $\mathbf{569.202 \pm 1.954}$ | $-2.83e + 05 \pm 3.99e + 04$ | $-959.169 \pm 342.340$ | $562.509 \pm 7.984$ | $-2.06e + 04 \pm 6.17e + 03$ |
| VDPB-r | $\mathbf{564.488 \pm 3.558}$ | $-2.66e + 04 \pm 2.04e + 04$ | $236.299 \pm 138.579$ | $552.442 \pm 7.922$ | $-3285.368 \pm 2764.594$ |
| VDPB-c | $\mathbf{574.775 \pm 4.241}$ | $-5.02e + 05 \pm 6.57e + 04$ | $-1749.567 \pm 575.022$ | $565.238 \pm 5.630$ | $-4.08e + 04 \pm 9.41e + 03$ |
| ES | $-13.032 \pm 0.910$ | $-13.541 \pm 1.204$ | $-13.687 \pm 1.294$ | $-13.533 \pm 1.201$ | $\mathbf{-12.876 \pm 0.937}$ |
| Lin-4 | $\mathbf{5.780 \pm 0.495}$ | $-26.195 \pm 8.401$ | $5.739 \pm 0.518$ | $-0.020 \pm 2.165$ | $5.763 \pm 0.553$ |
| Lin-8 | $\mathbf{89.049 \pm 2.772}$ | $-1387.313 \pm 201.479$ | $77.042 \pm 3.843$ | $-0.475 \pm 25.415$ | $88.506 \pm 2.930$ |
| Tanh-4 | $17.386 \pm 1.111$ | $-24.730 \pm 13.715$ | $\mathbf{17.404 \pm 1.081}$ | $12.408 \pm 3.369$ | $17.316 \pm 1.150$ |
| Tanh-8 | $\mathbf{311.528 \pm 3.827}$ | $-2941.246 \pm 364.753$ | $255.127 \pm 9.701$ | $239.432 \pm 17.222$ | $304.161 \pm 5.220$ |

Table 14: Results in variational inference for additional models: a non gated EMF-T, and the combination of Mean Field and Multivariate Normal with EMF-T and GEMF-T.

| | | EMF-T | MF-EMF-T | MVN-EMF-T | MF-GEMF-T | MVN-GEMF-T |
|---|---|---|---|---|---|---|
| BRS-r | -ELBO | $-3.293 \pm 0.974$ | $13.905 \pm 1.224$ | $-3.123 \pm 0.968$ | $-3.087 \pm 0.967$ | $-3.077 \pm 0.966$ |
| | FKL | $37.483 \pm 1.067$ | $-6.429 \pm 5.058$ | $32.458 \pm 0.775$ | $32.261 \pm 0.779$ | $32.351 \pm 0.822$ |
| BRS-c | -ELBO | $15.894 \pm 1.406$ | $18.602 \pm 1.723$ | $15.917 \pm 1.397$ | $15.960 \pm 1.399$ | $15.921 \pm 1.403$ |
| | FKL | $28.554 \pm 0.907$ | $27.117 \pm 1.373$ | $28.562 \pm 0.906$ | $28.500 \pm 0.907$ | $28.386 \pm 0.938$ |
| BRB-r | -ELBO | $-3.096 \pm 0.971$ | $13.905 \pm 1.224$ | $-3.123 \pm 0.968$ | $-3.150 \pm 0.967$ | $-3.077 \pm 0.966$ |
| | FKL | $32.291 \pm 0.856$ | $-6.429 \pm 5.058$ | $32.458 \pm 0.775$ | $32.426 \pm 0.764$ | $32.351 \pm 0.822$ |
| BRB-c | -ELBO | $11.878 \pm 0.990$ | $13.838 \pm 1.251$ | $11.912 \pm 0.985$ | $11.940 \pm 0.988$ | $11.930 \pm 0.988$ |
| | FKL | $27.002 \pm 1.885$ | $25.407 \pm 2.120$ | $26.880 \pm 1.914$ | $26.674 \pm 2.041$ | $26.845 \pm 1.924$ |
| LZS-r | -ELBO | $46.995 \pm 0.450$ | $527.145 \pm 319.063$ | $1255.087 \pm 330.588$ | $52.865 \pm 0.694$ | $1254.725 \pm 330.517$ |
| | FKL | $245.505 \pm 2.401$ | $-8660.092 \pm 5474.940$ | $-4.8e+04 \pm 1.9e+04$ | $-1976.326 \pm 2022.022$ | $-4.6e+04 \pm 1.8e+04$ |
| LZS-c | -ELBO | $8.290 \pm 1.031$ | $56.846 \pm 45.443$ | $23.895 \pm 1.942$ | $10.844 \pm 1.104$ | $23.540 \pm 1.881$ |
| | FKL | $245.214 \pm 1.583$ | $176.425 \pm 18.073$ | $-3.4e+04 \pm 9.4e+03$ | $192.223 \pm 16.425$ | $-3.7e+04 \pm 1.0e+04$ |
| LZB-r | -ELBO | $85.425 \pm 36.684$ | $187.988 \pm 87.731$ | $487.722 \pm 166.150$ | $148.388 \pm 57.199$ | $487.486 \pm 166.134$ |
| | FKL | $192.466 \pm 41.929$ | $-1.8e+04 \pm 1.1e+04$ | $-5.8e+04 \pm 2.8e+04$ | $-1.5e+04 \pm 1.4e+04$ | $-5.8e+04 \pm 2.8e+04$ |
| LZB-c | -ELBO | $5.800 \pm 0.479$ | $39.506 \pm 15.506$ | $17.014 \pm 1.592$ | $51.010 \pm 18.540$ | $16.732 \pm 1.557$ |
| | FKL | $249.858 \pm 1.379$ | $-2045.551 \pm 2165.953$ | $-4.4e+04 \pm 9.2e+03$ | $114.907 \pm 46.043$ | $-4.4e+04 \pm 9.2e+03$ |
| VDPS-r | -ELBO | $93.523 \pm 2.734$ | $102.648 \pm 3.052$ | $94.538 \pm 2.741$ | $94.467 \pm 2.748$ | $94.623 \pm 2.725$ |
| | FKL | $577.918 \pm 4.143$ | $567.666 \pm 6.758$ | $576.639 \pm 4.191$ | $577.257 \pm 4.164$ | $578.076 \pm 4.129$ |
| VDPS-c | -ELBO | $68.385 \pm 2.599$ | $69.348 \pm 2.656$ | $69.594 \pm 2.640$ | $68.873 \pm 2.574$ | $69.600 \pm 2.592$ |
| | FKL | $569.190 \pm 1.954$ | $568.875 \pm 1.962$ | $567.766 \pm 2.192$ | $568.860 \pm 1.945$ | $568.492 \pm 2.398$ |
| VDPB-r | -ELBO | $68.036 \pm 1.312$ | $75.147 \pm 1.387$ | $67.914 \pm 1.387$ | $68.980 \pm 1.314$ | $69.144 \pm 1.289$ |
| | FKL | $564.516 \pm 3.620$ | $557.422 \pm 3.521$ | $562.322 \pm 3.857$ | $562.386 \pm 3.693$ | $563.441 \pm 3.847$ |
| VDPB-c | -ELBO | $43.314 \pm 2.043$ | $43.783 \pm 2.096$ | $44.487 \pm 2.083$ | $43.680 \pm 2.070$ | $44.443 \pm 2.065$ |
| | FKL | $574.780 \pm 4.207$ | $574.330 \pm 4.057$ | $574.061 \pm 4.254$ | $574.561 \pm 4.170$ | $573.262 \pm 4.676$ |
| ES | -ELBO | $36.139 \pm 0.004$ | $36.794 \pm 0.040$ | $36.532 \pm 0.016$ | $36.798 \pm 0.041$ | $36.512 \pm 0.019$ |
| | FKL | $-13.043 \pm 0.906$ | $-13.525 \pm 1.203$ | $-13.666 \pm 1.295$ | $-13.582 \pm 1.219$ | $-13.732 \pm 1.311$ |
| Lin-4 | -ELBO | $1.512 \pm 0.573$ | $5.981 \pm 0.520$ | $1.513 \pm 0.574$ | $3.371 \pm 0.650$ | $1.533 \pm 0.572$ |
| | FKL | $5.806 \pm 0.518$ | $1.085 \pm 1.070$ | $5.842 \pm 0.499$ | $0.002 \pm 2.137$ | $5.821 \pm 0.528$ |
| Lin-8 | -ELBO | $2.609 \pm 0.208$ | $109.551 \pm 3.608$ | $5.600 \pm 0.260$ | $26.396 \pm 0.329$ | $4.173 \pm 0.195$ |
| | FKL | $89.005 \pm 2.776$ | $22.826 \pm 7.270$ | $87.689 \pm 3.056$ | $-2.279 \pm 26.566$ | $88.143 \pm 2.818$ |
| Tanh-4 | -ELBO | $-0.037 \pm 0.119$ | $6.221 \pm 0.161$ | $-0.025 \pm 0.125$ | $2.876 \pm 0.110$ | $-0.002 \pm 0.120$ |
| | FKL | $17.436 \pm 1.097$ | $6.913 \pm 4.124$ | $17.472 \pm 1.069$ | $12.561 \pm 3.256$ | $17.244 \pm 1.084$ |
| Tanh-8 | -ELBO | $1.866 \pm 0.119$ | $19.266 \pm 1.790$ | $5.338 \pm 0.124$ | $14.134 \pm 0.784$ | $4.237 \pm 0.194$ |
| | FKL | $311.594 \pm 3.618$ | $295.872 \pm 4.505$ | $285.647 \pm 7.425$ | $280.296 \pm 9.743$ | $303.473 \pm 5.529$ |

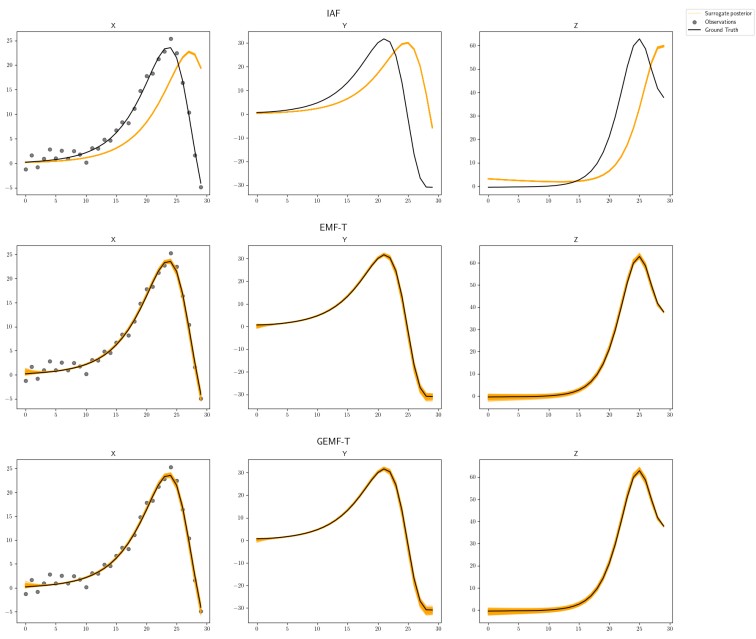

Figure 6: Surrogate posterior for Lorenz Smoothing.

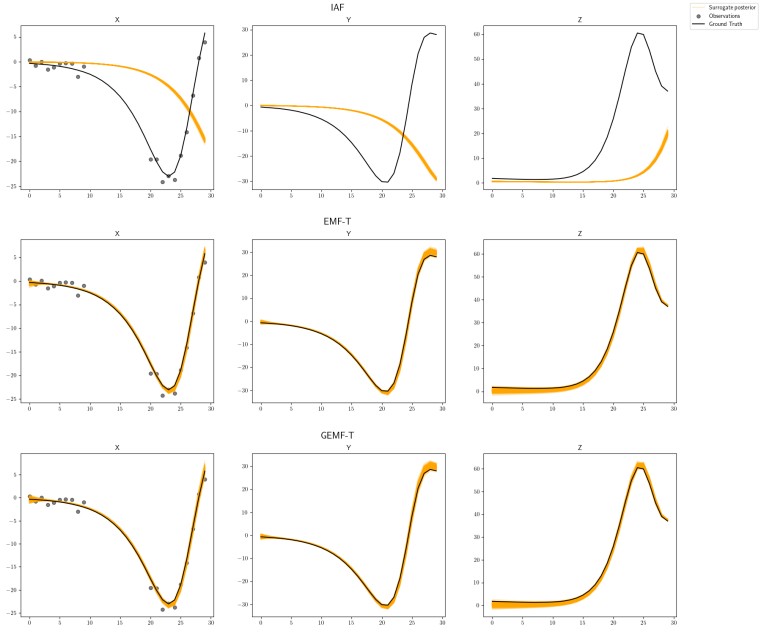

Figure 7: Surrogate posterior for Lorenz Bridge.

### A.7.6 SURROGATE POSTERIORS

We show and compare the obtained surrogate posteriors using IAF, EMF-T and GEMF-T, together with the ground truth and the observations (figures 6, 7, 8, 9).

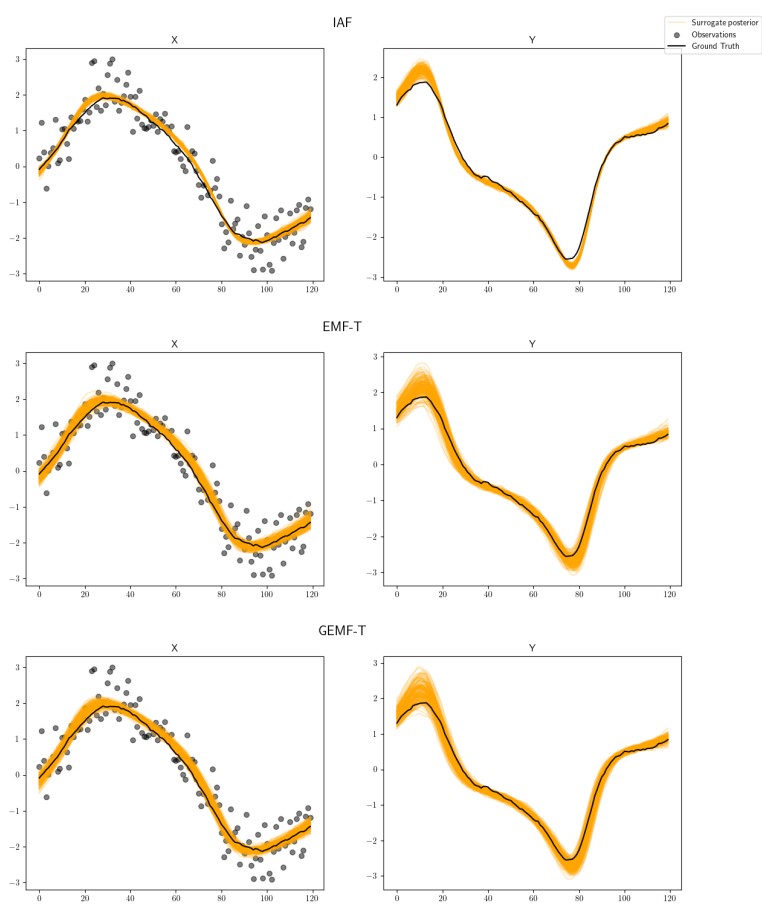

Figure 8: Surrogate posterior for Van der Pol Smoothing.

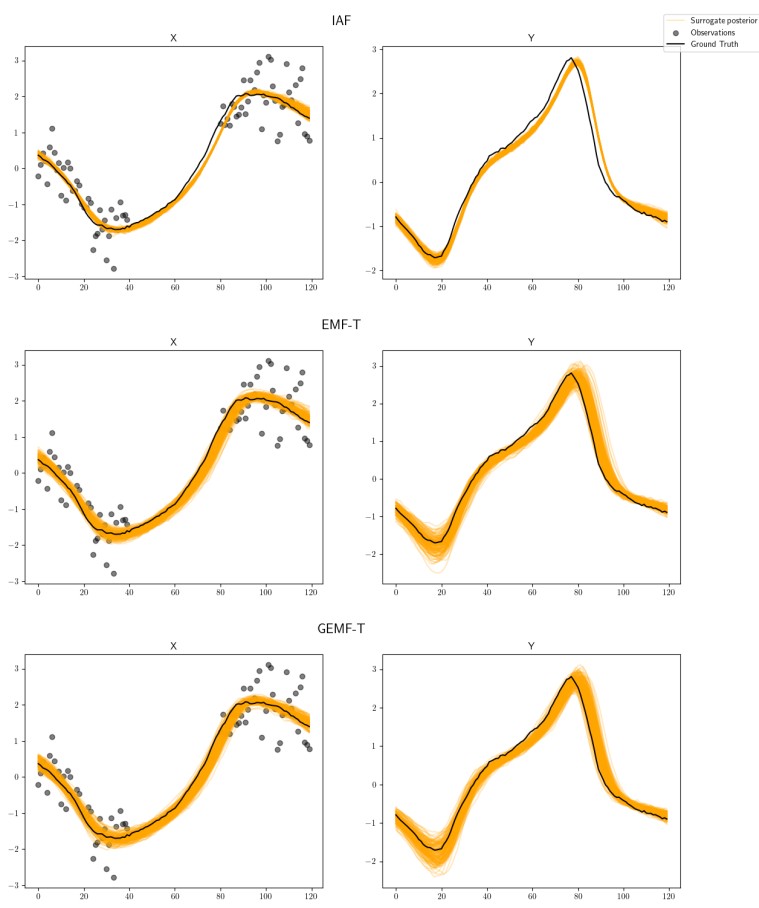

Figure 9: Surrogate posterior for Van der Pol Bridge.

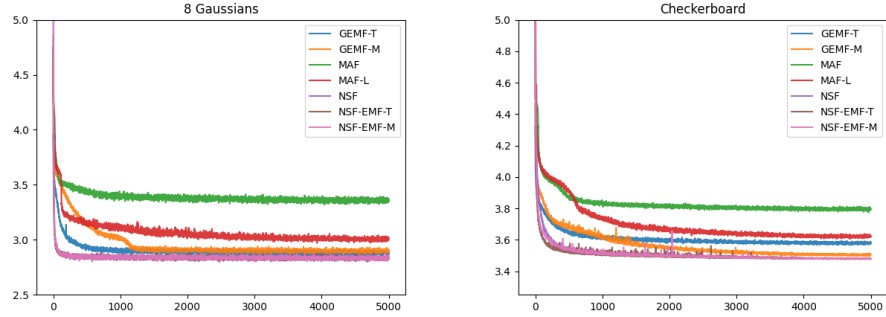

Figure 10: Training losses on 2D toy data. Each plotted value is the average loss of 100 iterations.

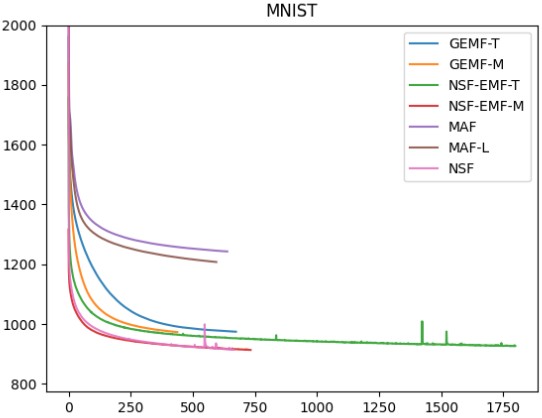

Figure 11: Training losses on MNIST. Some models trained for a smaller amount of epochs as they would overfit quicker on the validation set.

### A.7.7 TRAINING LOSSES

We show a comparison of the training losses for the different models, for the 2D data (figure 10, MMIST (figure 11, the IRIS and digits datsests (figure 13) and for the time series data (figure 12).

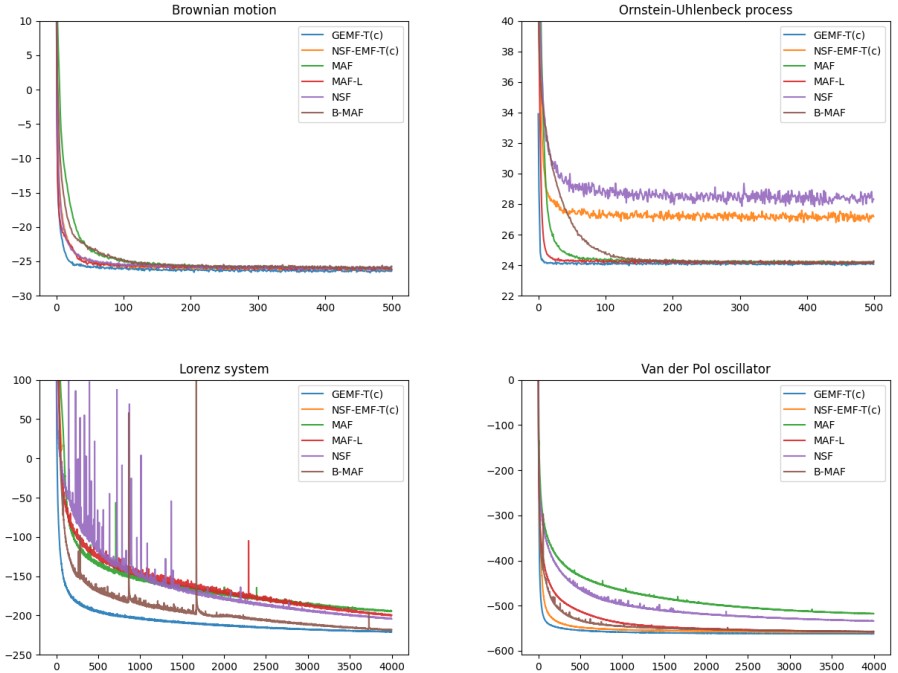

Figure 12: Training losses on time series toy data. Each plotted value is the average loss of 100 iterations.

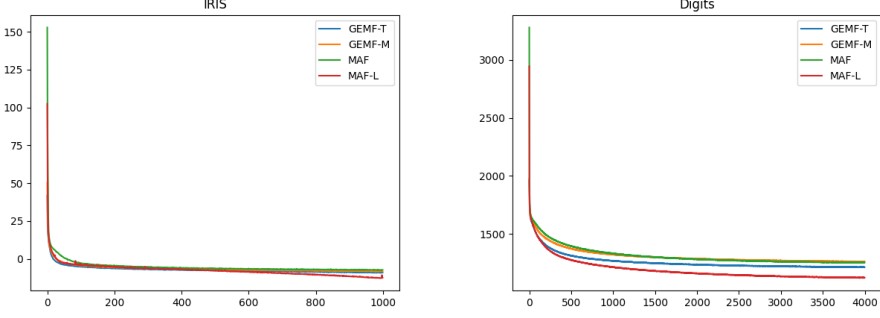

Figure 13: Training losses on Iris (left) and Digits (right) datasets. Each plotted value is the average loss of 100 iterations.

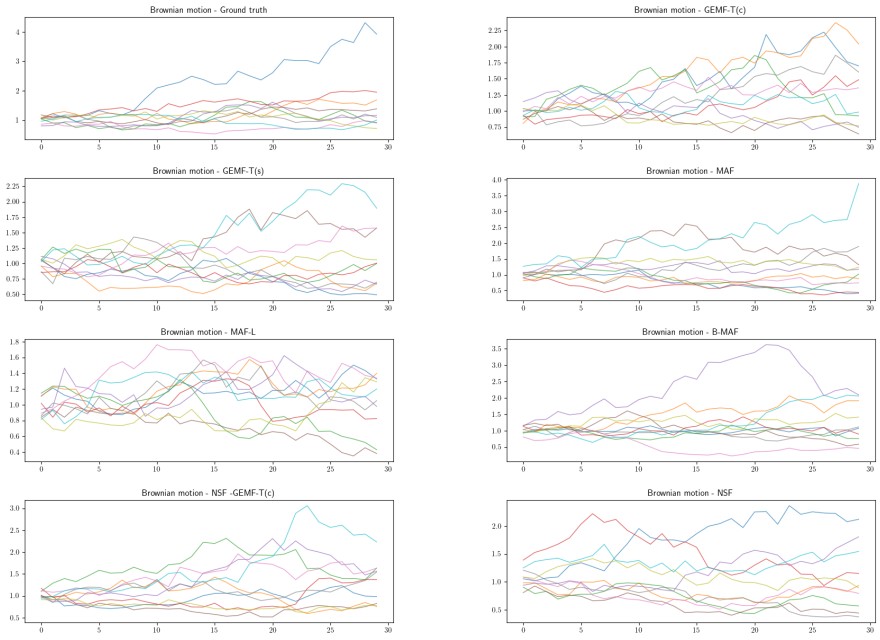

Figure 14: Samples for Brownian motion.

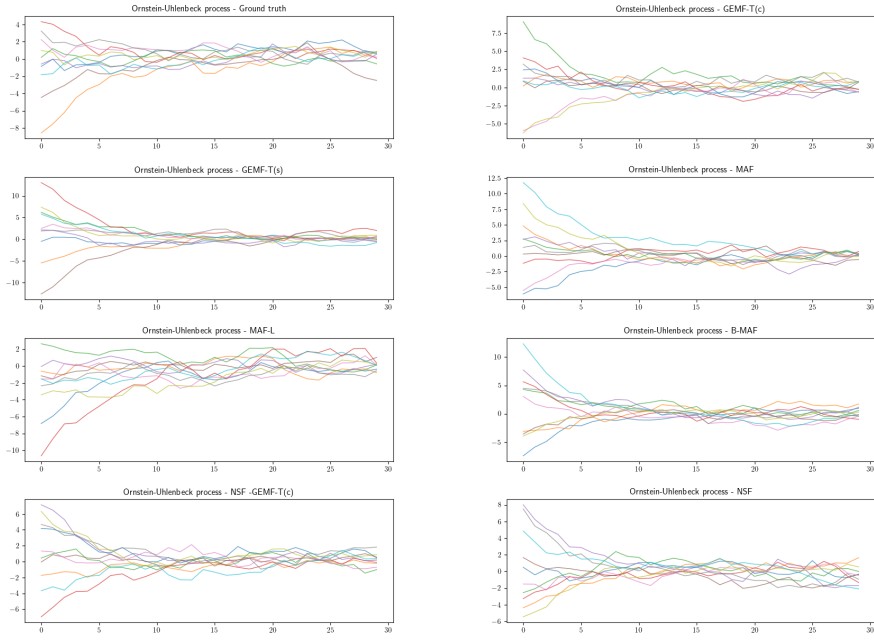

Figure 15: Samples for Ornstein-Uhlenbeck process.

### A.7.8 SAMPLES

We provide a comparison between the training data and the samples from the trained models. Figures 14 15 and 16 show samples for the Brownian motion, Ornstein-Uhlenbeck process and Lorenz system respectively.

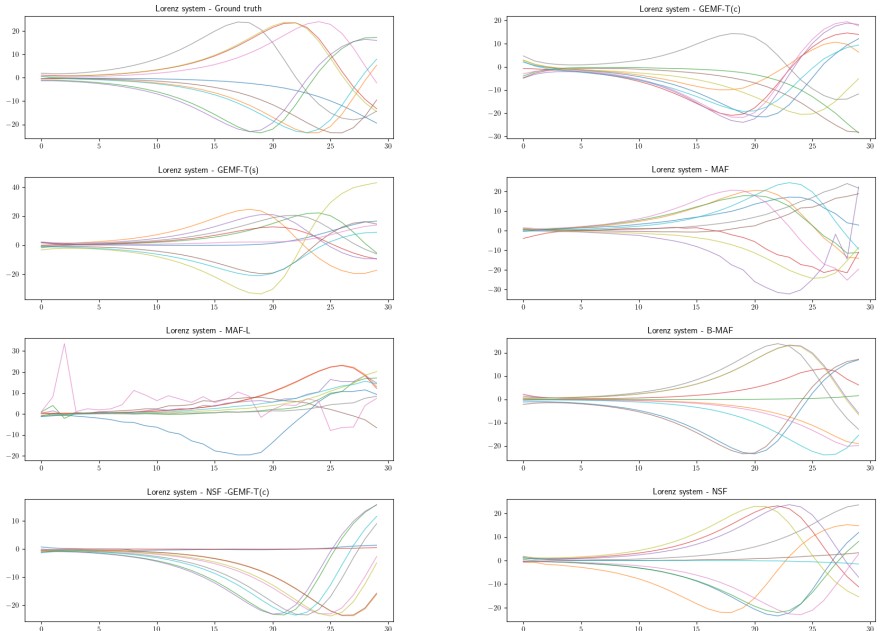

Figure 16: Samples for Lorenz system.

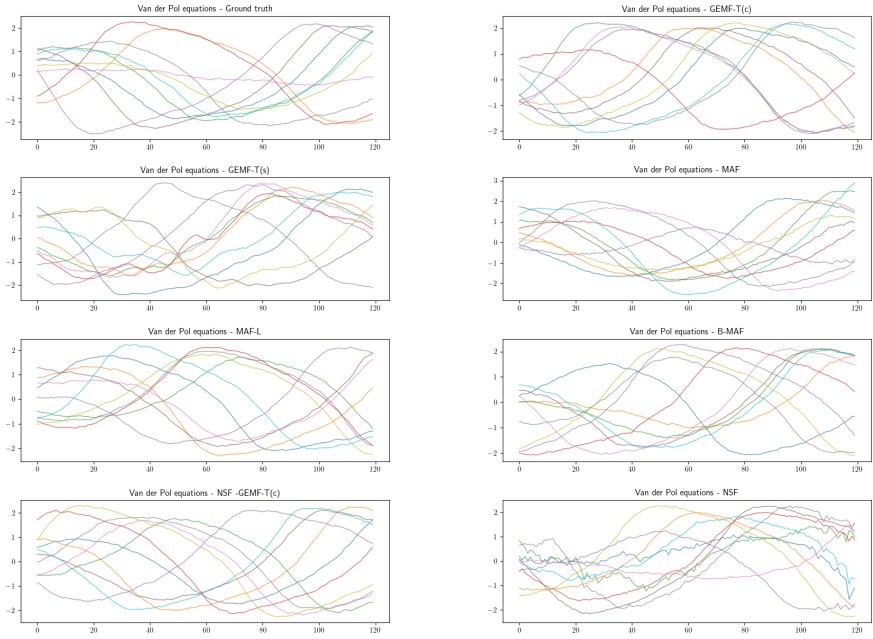

Figure 17: Samples for Van der Pol oscillator.

