# OpenReview forum: "Embedded-model flows: Combining the inductive biases of model-free deep learning and explicit probabilistic modeling"
_ICLR.cc/2022/Conference — ICLR 2022 Poster_

### Official Review · Reviewer_2pw8 · 2021-10-27

**Correctness:** 4
**Technical Novelty And Significance:** 3
**Empirical Novelty And Significance:** 3
**Recommendation:** 8
**Confidence:** 4

**Main Review:**

### Novelty
Although it is well-known that a probabilistic program (with deterministic control flow and continuous random variables) is reparameterisable with samples from a unit Gaussian, I believe that using this insight to construct a normalising flow layer is a novel and interesting idea. The "gated" layers introduced add further novelty (while being demonstrably useful). I am therefore confident that this paper makes sufficiently novel contributions.

### Strengths
- Incorporating the known structure/properties of complex distributions into usually black-box normalising flow architectures is a well-studied problem, which I suspect is important for many real-world applications. This paper presents a general solution applicable whenever a probabilistic model for the distribution can be written, even in the common case where it is only approximately correct (or e.g. only the prior is known but we are modelling a posterior).
- Experiments demonstrate the method on various distribution-modelling tasks, as well as its use for variational inference. Results are generally positive for the proposed method.
- The paper is mostly well-written.

### Weaknesses
- The experiments are all fairly small-scale with in the order of 10s of random variables. It would be interesting to see larger scale applications but I don't think this is necessary for publication.
- The method has some limitations (correct me if I'm wrong): (1) All variables must be continuous. (2) The probabilistic program must sample the same random variables on every execution. (3) The probabilistic program must be executed whenever the normalising flow is run (which could be a problem if the probabilistic program is slow to execute). There are not critical issues (and are shared by much related work), but are probably worth mentioning in the discussion.

### Minor
- In Section 5.2, it is misleading to say that "GEMF-T outperforms the baselines on all the problems", since the baselines obtain the best performance on two of the results in Table 5 in the appendix.
- Confusing equation: my understanding from Section 3.2 is that you set $x_{t+1} := f_{t+1, \sigma}(\epsilon_{t+1}; x_t)$. Why do you replace $\epsilon$ with $x_{t+1}$ in $f_{t+1, \sigma}(x_{t+1}; x_t) = x_t + \sigma x_{t+1}$ on the line below equation 13?

**Summary Of The Paper:**

The authors describe how to translate a probabilistic program into a normalising flow layer which maps samples from a unit Gaussian into samples from the probabilistic program. They propose a "gating" mechanism for this layer so that it can learn to interpolate between this transformation and the identity transform. They then show how this layer can be combined with generic normalizing flow layers, and in this way add an inductive bias to a normalising flow without sacrificing its expressivity. In the experiments, they provide various examples using probabilistic programs in this way to construct normalising flows with helpful inductive biases.

**Summary Of The Review:**

I'm recommending acceptance because this paper makes a novel, yet simple and generic, way to impose inductive biases on normalising flows. It is validated by experiments in various domains.

---

> ### Author Response · Authors · 2021-11-16
> **Response to Reviewer 2pw8**
>
> We thank the reviewer for the insightful and supportive review. We fixed the minor problems pointed out in the review and updated the manuscript accordingly. In the following, we address the main issues raised by the reviewer.
>
> ## Small scale experiments
>
> *“The experiments are all fairly small-scale within the order of 10s of random variables. It would be interesting to see larger scale applications but I don't think this is necessary for publication.”*
>
> Response: Given the generality of our approach, we decided to test several different settings instead of focusing our energy on one big problem. We think that this was appropriate as we needed to show the effectiveness of the general approach. We plan to perform larger scale experiments in follow up papers in collaboration with domain experts.
>
> ## Limitations
>
> *“The method has some limitations (correct me if I'm wrong): (1) All variables must be continuous. (2) The probabilistic program must sample the same random variables on every execution. (3) The probabilistic program must be executed whenever the normalising flow is run (which could be a problem if the probabilistic program is slow to execute). There are not critical issues (and are shared by much related work), but are probably worth mentioning in the discussion.”*
>
> Response: Correct. We included a larger treatment of those issues in the updated discussion section of the manuscript.

---

> > ### Comment · Reviewer_2pw8 · 2021-11-21
> > **Post-rebuttal**
> >
> > Thank you for your responses, both to my review and the others. After reading them, I'm happy to keep my rating.

---

### Official Review · Reviewer_zyZ8 · 2021-11-02

**Correctness:** 3
**Technical Novelty And Significance:** 3
**Empirical Novelty And Significance:** 3
**Recommendation:** 8
**Confidence:** 4

**Main Review:**

**Strengths:**
- Proposes a generic method to compile probabilistic programs into invertible functions given an ordering of the variables from the probabilistic program.

- Introduces a gating layer that allows the model to select whether to use the "prior" or not.  This is essentially a neural architecture search idea where the model is free to choose the prior or not.

- Shows better performance on structured VAE problems (particularly for the Lorenz system).

- Paper will provide code to actually do the compilation automatically from a probabilistic program. This could be generally useful to the community.

**Weaknesses:**
- There seems to be a disconnect between the probabilistic program (i.e., the domain knowledge) and flows.  As soon as the flow parts (i.e., MAF) modify the input to the EMF layer, the EMF layer no longer represents the probabilistic program, but rather than some distorted version of the probabilistic program. The gated part allows the model to completely ignore the prior if better. Thus, a key question remains: Why not just use a more state-of-the-art flow which can handle more arbitrary distributions? Why go through the trouble of carefully specifying a probabilistic model?

- Limited novelty in compilation approach: The idea of using conditional CDF functions is generally known as the "Rosenblatt transform".  Thus the construction of an invertible transformation to or from a Gaussian distribution via conditional CDF functions is not particularly novel.
[1] M. Rosenblatt, "Remarks on a multivariate transformation," Annals of Mathematical Statistics, vol. 23, no.3, pp. 470-472, 1952.

- The empirical results seem relatively weak (especially because the baselines seem weak).  It is unclear that the added work of incorporating domain knowledge significantly improves performance compared to just using more state-of-the-art flows. More details below.

    - Why not just use a more state-of-the-art normalizing flow (Neural Spline Flow, Residual Flow, Flow++, etc.)?  It is unclear what is the real advantage of EMF layers.  It seems that they add some modeling power but this extra modeling power could be added in different ways.

    - The multimodal part can be easily captured by Neural Spline Flows or more recent flow models than MAF as they have complex non-linear transformations that can model multi-modality.  Thus, it is unclear that the "multimodality" introduced by the EMF is caused by the "prior knowledge" about multimodality or merely because MAF is a very limited model and EMF is more complex. This would be significantly more convincing if this "prior knowledge" could be incorporated into models that already can model multimodality.  Overall, the experimental setup seems unfair as EMF introduces complex non-linearities that are unavailable to MAF (even with more parameters).

    - In the hierarchical experiment, again, it seems that even though the experimental setup favors hierarchical distributions, the MAF with more parameters does better.  Again, it's not clear the added benefit of EMF layers since they are not directly comparable to MAF (which has natural limitations).

    - In all these experiments, it seems that EMF should be the clear winner as the EMF incorporates domain knowledge that the baselines do not (i.e., it should have an unfair advantage because it uses external domain knowledge).  Yet, except for the Lorenz experiments, other simple baselines seem quite close.

- Limitation: The autoregressive transformation can be quite slow to compute in one direction.  This is why coupling layers have become quite useful in comparison to autoregressive invertible transformations.  The construction seems to inherit this computational issue with an autoregressive structure---i.e., if you want to use structured layers, you will be limited computationally in one direction.

**Other comments or questions**
- The gated architecture is useful for model power as the model can choose not to use the prior specification (i.e., the "bad" model).  However, this seems to be in partial opposition to the idea of incorporating prior knowledge---i.e., the goal of incorporating prior knowledge is to improve performance by biasing the model towards more appropriate solutions.  Thus, can these domain-specific or prior-specific models just be considered "hints" to help the algorithm find the best fitting model?

- Could the interpretability of the model be better because of these structured layers?  This could be an added benefit but it is unclear if that is true if the input and output of the structured layer can be arbitrarily manipulated by unstructured flow layers.

- Why was the dataset augmented in the hierarchical experiment?  Shouldn't the hierarchical EMF layer be able to learn the hierarchy automatically from the data?

- The Lorenz system seems to show the most improvement.  The BR and OU time series only show a marginal improvement of negative log likelihood.  Why is that?

- Regarding time series experiment, while the base distribution of B-MAF follows the continuity equation, why not also make the masked flow layers obey this ordering of the variables (as I believe MAF generally assumes a random order)? This would be a simple and straightforward way to incorporate the ordering of variables in the flow part of MAF.


**Summary Of The Paper:**

This paper first proposes to take a probabilistic program and compile it into a sequence of invertible transformations. Applying this invertible transformation to a standard normal distribution produces a sample from the probabilistic program.
This "compiled" invertible transformation is then used as a layer with learnable parameters, called a structured layer, in a larger flow model which can also have unstructured layers.
The paper argues that this enables the ability to incorporate domain knowledge via these structured layers while enabling deviation from this domain knowledge via the unstructured flow layers.
The paper performs experiments on multimodal, hierarchical, time series and variational inference data.


**Summary Of The Review:**

**Update since author response**:
I thank the authors for their very thoughtful response.  I greatly appreciated the new experiments and more discussion on why the structured layers are simple and have fixed parameters for essentially simple inductive biases.  Also, I appreciate the new NSF experiments.  Overall, I felt these responses answered most of my major concerns.   I have updated my score accordingly.  I would encourage the authors to incorporate some of this discussion in the paper if accepted.

----
Overall, I think the idea of the paper seems reasonable (i.e., to compile known domain structures into flows). However, the compilation technique using conditional CDFs is relatively well-known as the Rosenblatt transform, and it is unclear why a practitioner would not just want to use a more state-of-the-art flow instead of spending time to construct a probabilistic model.  The empirical results are unconvincing that a SOTA flow (even with similar number parameters) wouldn't outperform the current approach.

---

> ### Author Response · Authors · 2021-11-16
> **Response to Reviewer zyZ8 [1/3]**
>
> We kindly thank the reviewer for the very detailed and thoughtful review.
>
> As one of the main concerns was the strength of the empirical validation, we decided to run several additional more challenging experiments on a nonlinear dynamical model (van der Pol oscillator). We also included new neural spline flow EMF models and baseline to show that the improvements are maintained even when using more advanced generic architectures. The results of the new experiments are summarized in the response to all reviewers and show that our approach provides improvements even when integrated with more complex architectures such as neural spline flows.
>
> In the following, we will address the main issues raised by the reviewer.
>
> ## Disconnect
>
> *“ There seems to be a disconnect between the probabilistic program (i.e., the domain knowledge) and flows. As soon as the flow parts (i.e., MAF) modify the input to the EMF layer, the EMF layer no longer represents the probabilistic program, but rather than some distorted version of the probabilistic program. The gated part allows the model to completely ignore the prior if better.*
>
>  Response: We do not consider this as a weakness. **The fact that the generic flow layers can modulate the statistics and coupling of the original probabilistic program is fundamental to our approach as no human specified model is entirely correct and will include some miss-specification.** Most models do not account for all possible sources of couplings between variables. For example, the stochastic models used in finance are usually Markov and cannot account for long-memory effects. In this context, the generic layers can preserve the general structure and add the additional couplings. In our opinion, there isn't a disconnect but instead a synergy between the model-based and model-free parts of the architecture. **We support the claim empirically, as our experiments show that training does not destroy the model-specific bias such as continuity, multimodality or chaotic dynamics but instead it steers the model so as to better fit the data. This can be seen in the generated samples plotted in the supplementary.**
>
> ## Why should you use models instead of more complex generic architectures
>
> *Thus, a key question remains: Why not just use a more state-of-the-art flow which can handle more arbitrary distributions? Why go through the trouble of carefully specifying a probabilistic model?”*
>
> Response:
>
> We provided an extensive answer to this central question in the general response to all reviewers.
>
> ## Hard structure vs inductive bias
>
> *“The gated architecture is useful for model power as the model can choose not to use the prior specification (i.e., the "bad" model). However, this seems to be in partial opposition to the idea of incorporating prior knowledge---i.e., the goal of incorporating prior knowledge is to improve performance by biasing the model towards more appropriate solutions. Thus, can these domain-specific or prior-specific models just be considered "hints" to help the algorithm find the best fitting model?”*
>
> Response: Exactly. This is a very important point and central to our work. Indeed, the model structure does not hard constraint the expressivity of the architecture, which can still represent any distribution. However, the resulting inductive bias is very different as, if the model is chosen appropriately, training will usually converge to a local minimum that can be interpreted as a perturbation of the model. This was the case in all our experiments.
>
> ## Novelty
>
> *“Limited novelty in compilation approach: The idea of using conditional CDF functions is generally known as the "Rosenblatt transform". Thus the construction of an invertible transformation to or from a Gaussian distribution via conditional CDF functions is not particularly novel. [1] M. Rosenblatt, "Remarks on a multivariate transformation," Annals of Mathematical Statistics, vol. 23, no.3, pp. 470-472, 1952.”*
>
> Response: We thank the reviewer for providing the reference, we are going to add the reference to Rosenblatt's original paper. However, we do not think that this significantly reduces the novelty of our approach for three reasons:
>
> - The main contribution of the paper is in the context of the normalizing flow literature. Our approach for embedding domain knowledge to flow architectures is conceptually new.
> - Our layers include local gates that are not present in the original transformation.
> - Using EMFs, we introduced a novel generic and fully automated variational inference method that exploits the structure of the prior and outperforms most existing structured variational inference solutions.

---

> > ### Author Response · Authors · 2021-11-16
> > **Response to Reviewer zyZ8 [2/3]**
> >
> > ## Empirical validation
> >
> > *“The empirical results seem relatively weak (especially because the baselines seem weak). It is unclear that the added work of incorporating domain knowledge significantly improves performance compared to just using more state-of-the-art flows. More details below. Why not just use a more state-of-the-art normalizing flow (Neural Spline Flow, Residual Flow, Flow++, etc.)? It is unclear what is the real advantage of EMF layers. It seems that they add some modeling power but this extra modeling power could be added in different ways.”*
> >
> > Response: First of all, we would like to clarify an important point. **The main aim of EMF is not to add modeling power. Our structured layers can have very few parameters and do not necessarily increase the expressivity of the network.** For example, the continuity and hierarchical models are linear without trainable parameters and do not increase expressivity. Instead, the structured layer introduces strong inductive biases so that the network converges to minima that represent "perturbations" of the original model. If the model captures important features of the data, this leads to better performance without the need for higher expressivity.
> >
> > In the initial submission, we decided to use MAF as generic flow layers since they are well-tested, easy to calibrate, computationally efficient and have tested open-source implementations available. We agree that using more complex architectures would be informative and we are currently running a series of new experiments. Therefore, in the updated manuscript we included both Neural Spline Flows (NSF) EMF models and baselines to the multimodal and timeseries experiments. All results are consistent with our original experiments. This shows that the improvements provided by our approach cannot be easily achieved by adding extra complexity.
> >
> > ## Multimodality experiment
> >
> > *“The multimodal part can be easily captured by Neural Spline Flows or more recent flow models than MAF as they have complex non-linear transformations that can model multi-modality. Thus, it is unclear that the "multimodality" introduced by the EMF is caused by the "prior knowledge" about multimodality or merely because MAF is a very limited model and EMF is more complex. This would be significantly more convincing if this "prior knowledge" could be incorporated into models that already can model multimodality. Overall, the experimental setup seems unfair as EMF introduces complex non-linearities that are unavailable to MAF (even with more parameters).”*
> >
> > Response: Neural spline models are explicitly designed to deal with multimodality. As their inductive bias is already geared towards modeling multimodality, we therefore do not think that it is particularly useful to add a multimodal structured layer in a Neural Splines Flow (NSF) architecture. On the other hand, we do agree that NSF is an important baseline that would help contextualize our work. We therefore included the NSF baseline in the updated manuscript. In the simple problems considered, our multimodal EMF architectures perform comparably with the NSF. For the sake of completeness, we also included the results of a NSF model with structured multimodal layer, which in fact achieved top performance (see Table 1 and figures 3 and 4).

---

> > > ### Author Response · Authors · 2021-11-16
> > > **Response to Reviewer zyZ8 [3/3]**
> > >
> > > ## Hierarchical, timeseries and variational experiments
> > >
> > > *“In the hierarchical experiment, again, it seems that even though the experimental setup favors hierarchical distributions, the MAF with more parameters does better. Again, it's not clear the added benefit of EMF layers since they are not directly comparable to MAF (which has natural limitations). In all these experiments, it seems that EMF should be the clear winner as the EMF incorporates domain knowledge that the baselines do not (i.e., it should have an unfair advantage because it uses external domain knowledge). Yet, except for the Lorenz experiments, other simple baselines seem quite close.”*
> > >
> > > Response: In the timeseries and hierarchical experiments we use extremely basic structured models, which correspond to sparse linear layers and without learnable parameters. These layers do not increase the expressivity of the networks. Nevertheless, this simple EMF tends to perform better than the large MAF while having fewer parameters (Tables 5, 6, 7 of Appendix A.5) and all the very same limitations of the original MAF architecture (linear layers do not add expressivity). The fact that such a simple linear layer can substantially improve performance even in spite of smaller depth and number of parameters is in our opinion remarkable. The better performance can be seen visually in the samples (see figures in Appendix A.7.4). As shown in the variational experiment, the performance boost becomes major when the model is appropriate and more complex. To strengthen this point, we run a new series of experiments with a new dynamical model (van der Pol oscillator) both in the time series and in the variational case. The results confirm the same pattern observed in the Loretz experiments.
> > >
> > > ## Questions
> > >
> > > ### Performance
> > >
> > > *“The Lorenz system seems to show the most improvement. The BR and OU time series only show a marginal improvement of negative log likelihood. Why is that?”*
> > >
> > > Response: Because BR and OU are simple models that in theory can be learned almost perfectly even by linear normalizing flows. We therefore decided to strengthen the empirical validation by adding an additional dynamical model (var der Pol oscillator).
> > >
> > > ### Continuity of baselines:
> > >
> > > *“Regarding time series experiment, while the base distribution of B-MAF follows the continuity equation, why not also make the masked flow layers obey this ordering of the variables (as I believe MAF generally assumes a random order)? This would be a simple and straightforward way to incorporate the ordering of variables in the flow part of MAF.”*
> > >
> > > Response: In the B-MAF baseline, we do not shuffle the MAF layers so as to preserve the ordering of the continuous process. We clarified this in the paper. The other MAF models were shuffled as we wanted to compare against models without user-defined inductive biases.

---

> > > > ### Comment · Reviewer_zyZ8 · 2021-11-20
> > > > **Limitation**
> > > >
> > > > Thank you for your responses, I have updated my review and score.  I did notice that you did not respond to the limitation mentioned.  Could you respond to the limitation mentioned?
> > > >
> > > > Limitation: The autoregressive transformation can be quite slow to compute in one direction. This is why coupling layers have become quite useful in comparison to autoregressive invertible transformations. The construction seems to inherit this computational issue with an autoregressive structure---i.e., if you want to use structured layers, you will be limited computationally in one direction.

---

> > > > > ### Author Response · Authors · 2021-11-21
> > > > > **Answer to Limitation**
> > > > >
> > > > > We have added sampling time for all the models in tables 8, 9, 10 of appendix A.5. There is indeed an increase in sampling time when the induced structure is autoregressive. However, combining a flow with coupling layers with an autoregressive structure has a much lower overhead than using a flow with autoregressive layers. We updated the discussion section, in which we now cover the sampling complexity issue.

---

> ### Comment · Reviewer_zyZ8 · 2021-11-24
> **Key followup: Ordering of variables**
>
> Hi authors,
>
> Thank you for your responses.  I had one additional comment.  How do you determine the ordering of variables when creating the autoregressive transformation?  This seems to be a critical question when compiling a program and will generate very different structured layers.  This is similar to variable elimination ordering.
>
> If I missed it in the original paper, please just point me to the section. If you have not discussed it already, please answer how you choose the variable ordering and why.  What are the potential pitfalls of different variable orderings?  Are there ways to mitigate these pitfalls (e.g., multiple structured layers with different orderings in parallel)?  Would it be best for the variable ordering to be as causal as possible?
>
> Looking forward to your response.

---

> > ### Author Response · Authors · 2021-11-24
> > **Response: Ordering of variables**
> >
> > In our use cases, the variables of the probabilistic programs are assumed to be associated with the indices of the observables, so that each node in the program corresponds to a dimension of the data. This is a common assumption to make in settings where explicit models are available since these models are fitted by maximum likelihood assuming this a-priori association. For example, in timeseries analysis the data is itself sorted by its timestamp and "aligned" to the model. In these cases, there is no ambiguity due to variable ordering. While it is true that there usually are multiple orderings compatible with the parenting structure (e.g. A > B if B is an ancestor of A), all these equivalent orderings result in the very same structured layer. In fact, the bijective transformations associated to a set of unordered variables can be applied in parallel. Our TFP implementation handles this sorting and parallelization automatically. We are going to include the details of the code in the supplementary.
> >
> > There can indeed be interesting situation where a model is available but the mapping of its variables to the dimensions of the data is not known a priori. In this case, indeed different mappings will result in different structured layers. In this latter case, which however does not fall in the range of problems we considered in this paper, it would be indeed interesting to use several structured layers with randomized orders.
> >
> > We hope this answer your question.

---

> > > ### Comment · Reviewer_zyZ8 · 2021-11-24
> > > **Ordering based on program**
> > >
> > > Thanks for the response! If I understand correctly, then this means that the probabilistic program inherently defines the ordering of variables (i.e., a partial ordering of variables given by the program).  This is essentially a directed acyclic graph that is created as the variables are defined in the program.  Thus, while two probabilistic programs could represent the same distribution, you assume that the way the model is defined by the programmer is the "correct" and likely "causal" ordering of variables.  (As an example, the joint distribution of a multivariate Gaussian can have any ordering of variables, x_1 => x_2 or x_2 => x_1 but the programmer will define this ordering based on domain knowledge.)

---

> > > > ### Author Response · Authors · 2021-11-24
> > > > **Response: Ordering based on program**
> > > >
> > > > Exactly.
> > > > Indeed causality is often a guiding principle when designing the model and the ordering of its connection. For example, in timeseries analysis the future values is usually assumed to depend on the past values (e.g. the autoregressive models used in econometrics, Euler discretized equations of motion from physics).
> > > > However, I would not go as far as saying that the modeler assumes to have the "correct" ordering. Ultimately the ordering is a design choice that can be driven by pragmatic reasons. For example, the user can "slice" a 2D spatial process in one selected dimension so to induce an ordering (this is often done by numeric PDE solvers). This is useful but it does not necessarily reflect an intrinsic property of the data.

---

### Official Review · Reviewer_ihM2 · 2021-11-05

**Correctness:** 3
**Technical Novelty And Significance:** 3
**Empirical Novelty And Significance:** 3
**Recommendation:** 6
**Confidence:** 4

**Main Review:**

I do not count this as a weakness of the paper (but it should be changed): It seems that the paper is using the ICLR 2021 (last year) format because the page headers say “Under review as a conference paper at ICLR 2021”.
### Strengths
* Incorporating inductive biases into flow architectures is a very interesting and important problem to deal with.
* The method presented in section 3.1 of converting a distribution into a uniform distribution and the uniform distribution into another one sounds very interesting and applicable in other settings.
* The presented model outperforms the baselines in the presented experiments.

### Weaknesses
* The paper claims outperforming the state-of-the-art. I do not think that this conclusion can be drawn from the presented experiments:
* The paper compares the suggested architecture with two flow models, Inverse Autoregressive Flows (IAFs) and Masked Autoregressive Flows (MAFs), which were published in 2016 and 2017, respectively. Both baseline models are dated and do not constitute the current state of the art. In a more recent survey on normalizing flows ([1], p. 13), MAFs did not achieve state-of-the-art performance on any of the covered datasets.
* The datasets that the paper uses for evaluation are not common for assessing the quality of normalizing flow models. None of the datasets is used in [1]. In particular, neither the IAF paper nor the MAF paper use these datasets for evaluation. I recommend evaluating the presented model on some (simple) image datasets, like MNIST and CIFAR10, which are very commonly used in the normalizing flow literature.
* The conclusion states “We showed how, by choosing appropriate inductive biases, EMF can improve over generic normalizing flows on a range of different domains, with only a negligible increase in complexity …”
This conclusion should only be drawn if it can be supported by some concrete numbers. To the best of my knowledge, the paper does not mention anywhere how the complexity of the proposed EMF compares to the complexity of the baseline model. Indeed, presenting training and sampling times of the proposed and the baselines models would strengthen the contribution.
Clarity
* I have really tried to understand the methodology in the main section 3.2, but the section is still not clear to me. I understand the univariate case in section 3.1, but I cannot make sense of the main section 3.2.
In Figure 2, I do not understand what expressions like model_generator, gen.send and d.as_bijector mean. Moreover, the function forward_and_log_det_jacobian seems to be called within itself, but with only one argument instead of the 3 used in the definition. (The same applies to inverse_and_log_det_jacobian.)
What is the input and the output of the structured layers? What do the epsilon variables in e.g. equation (5) mean? Shouldn’t the epsilons only come from the flow latent space and not be present in intermediate layers?
A toy example where each step of the proposed layers is explained for some easy problem would be really helpful. This could either be put into the appendix or replace Figure 1, which I did not find very illuminating.

**Minor and typos**
1. x_0 is not present on the left hand side of equation (2), so it should not be on the right hand side either.
2. On p. 3, in the sentence “The probability integral transform theorem states …”: The upper limit of the integral should be x and not y.
3. p. 2: “We call these architectures embedded-model flow s (EMF).” -> *flows*

[1] Normalizing Flows: An Introduction and Review of Current Methods, Kobyzev, Prince, Brubaker.


**Summary Of The Paper:**

This paper proposes a new type of normalizing flow for incorporating inductive biases into the model architecture. To this end, the authors introduce a so-called “structure layer”, aiming to transform a spherical Gaussian variable into a pre-defined probabilistic program. A modified version, called “gated structured layer” is proposed in order to skip problematic parts of the model. The model is evaluated on a variety of different datasets and outperforms the masked autoregressive flow baseline.

**Summary Of The Review:**

Incorporating inductive biases into normalizing flow architectures is a very relevant and important problem. Some of the techniques presented in the paper seem interesting, but I really struggled to understand the main methodology part in the way it was presented. The paper claims to outperform the state-of-the-art normalizing flows. For the reasons stated above, I do not agree with this. The paper would benefit from further experiments on more common datasets and from a comparison with more recent state-of-the-art flow models.

---

> ### Author Response · Authors · 2021-11-16
> **Response to Reviewer ihM2**
>
> We kindly thank the reviewer for the detailed review. We are going to update the manuscript and correct all the identified minor mistakes, including the use of the older format.
>
> However, we think that some of the reasons given for deciding to reject are based on possible misunderstandings:
>
> - The "state-of-the-art" claim in the abstract and in the discussion was referred to the automatic structured variational inference prior-embedding method, where we tested on the set of structured problems used in [1] and [2]. We did not state that our networks beat state-of-the-art flows in generative modeling. In fact, we are not proposing the use of any particular architecture. The decision to use MAF layers both in EMF and baseline was motivated by simplicity and availability of well-tested implementations. So said, in order to avoid misunderstandings, we removed the state-of-the-art claim. We also included both a version of our method and a baselines with a more advanced neural spline architecture.
> - We think that the recommendation to validate the method on standard datasets such as MNIST and CIFAR10 is not appropriate in the context of our paper as our intent is to combine explicit domain knowledge with generic flow modeling. Natural images have a very complex structure that cannot be easily expressed with explicit models. The same argument can be made concerning the standard unstructured UCI datasets commonly used in the flow literature. While we agree that this results in an unconventional empirical section, we do think that it is unavoidable given the scope of the paper.
> - While we agree that the presentation can be improved, we do not think that the lack of clarity justifies rejection based on the fact that all other reviewers understood the main contribution and some praised the writing. We do however agree that the provided Python code was unintuitive. To make the paper clearer, we now included a more transparent pseudocode of forward transformation and Jacobian computation (Algorithm 1) and included a link to a Jupyter notebook tutorial (https://anonymous.4open.science/r/EmbeddedModelFlows-9172/EmbeddedModelFlows.ipynb) which works out the implementation in some simple models. Note that explicit treatment of several simple models is given in the experiments section.
>
>
>
> ## References
>
> [1] "Automatic structured variational inference." *International Conference on Artificial Intelligence and Statistics*. PMLR, 2021.
>
> [2]  "Automatic variational inference with cascading flows." *ICML* (2021).

---

> > ### Comment · Reviewer_ihM2 · 2021-11-16
> > **Regarding experiments on images**
> >
> > I thank the reviewers for their revisions and in particular for the Jupyter notebook. I am going to reread the paper, the code and the other reviews and reply within the next days. I'm happy to raise my score if the revised version is clear to me.
> >
> > **Regarding the validation on image datasets**
> > * Image datasets are often multimodal. The images in MNIST and CIFAR10 correspond to 10 classes, so it would be reasonable to assume that these datasets could be modeled with a mixture of 10 Gaussians. Particularly for MNIST the intra-class variation is not so high, so I imagine that the proposed layers could give good results there. Given that these datasets are so commonly used, I think that the experimental section would really benefit from them. If this experiment goes well, then this would show the benefits of the proposed method. If it did not go well, this would show the limitations of the proposed approach, which is very valuable information for readers.

---

> > > ### Author Response · Authors · 2021-11-16
> > > **Reply to: Regarding experiments on images**
> > >
> > > Dear reviewer,
> > >
> > > Thank you very much for the quick reply. We would be happy to add further clarifications to the paper and notebook if you think they are required to better understand our contribution.
> > >
> > > Concerning the image datasets, we agree that they make sense in the context of the multimodal experiments. The reason we decided to not have them in is that multimodality is a very simple and loose form of domain knowledge and we did not want to make it our main focus. All in all neural spline architectures are already very good at dealing with multi-modality and we are not claiming to offer the best multi-modal architecture. In our opinion, the importance of our contribution is mostly apparent when using structured programs such as the ones we use in timeseries analysis and especially variational inference.
> > >
> > > So said, we do agree that multi-modal experiments on MNIST and MNIST-like datasets can be informative and we are currently working on it. We are going to try to add them to the document by the end of the rebuttal period. However, given time and computation constraints we cannot promise that they will be ready in time.

---

> > > > ### Author Response · Authors · 2021-11-21
> > > > **Experiments on MNIST**
> > > >
> > > > We have added the MNIST results in section 5.1 The samples from the models can be found in figure 5, appendix A.7.2, and the training losses in figure 11, appendix A.7.6. The use of multimodality structure does improve over Masked Autoregressive Flows, and performs slightly worse than the Neural Splines Flow. We also see how, combining the multimodality structure with NSFs improves the performances, despite the fact that NSFs are already capable of modelling multimodalities.

---

> > > > > ### Comment · Reviewer_ihM2 · 2021-11-21
> > > > > **Reply**
> > > > >
> > > > > I thank the authors for the additional experiment.
> > > > > Most of my concerns are now resolved. The main algorithm is much clearer than the first version. I've raised both my score and my confidence. I find the main idea of the paper really intriguing and would have given an even higher score if  the proposed method was compared to some better Flow architectures. This is the main concern that is left for me.
> > > > >
> > > > > Now I understand the main section 3.2, but it took me quite a bit of time. I have some further suggestions improving the clarity of the paper:
> > > > > *  It was not clear to me at first that f_{j, \theta} corresponds to f_\theta from equation (4). It would be beneficial to formally define the function.
> > > > > * Throughout the paper, there is an ambiguity between a single layer and the entire Flow model. E.g. in algorithm 1, the input should not be called $\epsilon$ since this is already used for the normally distributed variables in the latent space of the flow. Similarly, the output should not be called $\mathbf{x}$, because that's the observed data points. I would suggest to let the input and the output of layer $i$ be called $\mathbf{h}_{i-1}$ and $\mathbf{h}_i$, respectively.
> > > > > * (Minor: It would make more sense to denote the Jacobian by $J_{j, \phi}$ instead of $J_{\phi, j}$ in order to aline the subscript with the function $f_{j, \phi}$.)

---

> > > > > > ### Author Response · Authors · 2021-11-21
> > > > > > **Reply and comments about SOTA flows**
> > > > > >
> > > > > > Dear reviewer,
> > > > > >
> > > > > > Thank you very much for the updated grade and the additional feedback. We agree that these modifications will increase the clarity of the paper and we are going to incorporate them in the camera-ready upon acceptance.
> > > > > >
> > > > > > Concerning the point
> > > > > >
> > > > > > *"would have given an even higher score if the proposed method was compared to some better Flow architectures"*
> > > > > >
> > > > > > We would like to stress that we did include rational quadratic neural spline flow baselines (NSF) in the updated manuscript. The results are described in the updated document, in the general response and in the response to reviewer zyZ8.
> > > > > > Neural spline flows achieve top results in [1] and we think are an appropriate state-of-the-art baseline for the kind of problems analyzed in our work.
> > > > > >
> > > > > > [1] Normalizing Flows: An Introduction and Review of Current Methods, Kobyzev, Prince, Brubaker.

---

### Official Review · Reviewer_g1yH · 2021-11-09

**Correctness:** 3
**Technical Novelty And Significance:** 3
**Empirical Novelty And Significance:** 3
**Recommendation:** 6
**Confidence:** 3

**Main Review:**

The strengths and weaknesses are pretty much discussed in the paper. Specifically,

The advantage of the method lies in the ability to control the flow model by injecting inductive bias. The whole model can be considered as a mixture of freely trainable MAF layers and fixed probabilistic programs, connected by the gated layers. The idea is novel and inspiring to me.  The whole idea looks natural, it could be considered as an augmentation of existing flow models that work well for small-sized data.

Several weaknesses that I've noticed:
1. The paper overall is easy to follow, however, I find the algorithm detail hard to read. Especially Figure 2, I encountered some difficulties in understanding the algorithm details as python. For instance, how as_bijector(gated=True) is implemented?
2. There are some typos in Section 3.1, when $C_{\phi}$ is introduced, it should be $\int_a^x$ right? and $a$ is undefined.
3. The gated layer requires root finding as mentioned, so exactly how much overhead is introduced? does this algorithm scalable to high dimensional data?
4. I noticed other methods such as Glow (Kingma et al.) not mentioned here, could authors explain why?

**Summary Of The Paper:**

This paper tries to bridge the concept of the normalizing flow model with the structured layers that encode the domain knowledge (also called inductive bias here). The general idea is to first notice that many existing probablistic programs (such as univariable RV) can be converted to flow under some special $f_{\phi}$.  Furthermore, the authors proposed a gated layer to allow the model to switch between user-specified model and the learned MAF. Extensive experiments in toy multimodality distributions, hierarchical gaussian, timeseries models and variational inferences are conducted to compare the proposed method with other baselines.



**Summary Of The Review:**

Overall, I think this paper advances the research in this area, although more careful ablations are required to ensure the empirical results are indeed promising.

---

> ### Author Response · Authors · 2021-11-16
> **Response to Reviewer g1yH**
>
> We thank the reviewer for the insightful and positive review. We took care of fixing all the minor mistakes in the updated manuscript. In the following, we will discuss the main concerns that have been raised.
>
> ## Clarity
>
> *“The paper overall is easy to follow, however, I find the algorithm details hard to read. Especially Figure 2, I encountered some difficulties in understanding the algorithm details as python. For instance, how asbijector(gated=True) is implemented?”*
>
> Response: We agree that the code in the paper was not very clear. We included a hopefully clearer pseudo-code in the updated manuscript (Algorithm 1). We also included the link to a repository in order to allow the reviewers and readers to understand the actual implementation (https://anonymous.4open.science/r/EmbeddedModelFlows-9172/README.md).
>
> ## Root finding overhead
>
> *“The gated layer requires root finding as mentioned, so exactly how much overhead is introduced? Is this algorithm scalable to high dimensional data?”*
>
> Response: When the conditional distributions are Gaussian, root finding is not needed as we can use a closed form formula. Root finding adds an overhead that is difficult to estimate in general since convergence depends on the form of the function. However, the algorithm is only used to invert univariate densities and is completely parallelized. Therefore, everything scales well as a function of dimensionality. To study the overhead, we included a table of sampling wall times for the 2D experiments in Table 10. In this experiment no overhead is present during training since only the inverse is needed in this regime.
>
> ## References
>
> *“I noticed other methods such as Glow (Kingma et al.) not mentioned here, could authors explain why?”*
>
> Response: We included the reference in the updated manuscript (see Section 1).

---

### Author Response · Authors · 2021-11-16
**General Response [1/2]**

We are grateful for your deep, helpful and insightful review. We are pleased that the reviewers appreciated the importance of the problem and the generality of our proposed solution. We also highly appreciated the numerous and well-thought suggestions to improve the paper. Based on these suggestions, we improved the manuscript in the following ways:

- **Empirical:** We straightened the empirical validation in the following ways:
  - We included a neural spline flow (NSF) baseline model in the multimodal experiments. We show that, at least in this simple setting, our multimodal EMF performs comparably with this state-of-the-art architecture. We also included experiments that integrate EMF with NSF layers, which achieve the best performance. The results can be found in Table 1 and figures 3 and 4.
  - We included an NSF-EMF version of our approach and a NSF baselines in the timeseries generative model. At the moment we only have results for the Lorentz model, we are going to add the remaining results to the manuscript when ready. The results in Table 3 confirm our previous experiments. The use of the continuity structured layer increases the performance of the NSF model similarly to our previous results with MAF architectures. Conversely to the multimodal experiment, NSF does not increase performance when compared to MAF in this setting. This is not very surprising as the transition models of the non-linear dynamics are conditionally Gaussian and therefore well captured by the MAF.
  - We added an additional analysis of dynamical systems (Van der Pol oscillator) both in the generative modeling and in the variational inference experiments. We find that EMF models outperform all baselines in all comparisons.

- **Clarity:** We increased clarity by replacing the Python code with a more transparent pseudocode (Algorithm 1). We also linked the paper to a (anonymous) repository containing the code and a short Jupyter notebook tutorial that showcases the implementation in a few simple settings (https://anonymous.4open.science/r/EmbeddedModelFlows-9172/README.md). We also included a more detailed explanation of the recursive formula in Eq.5. Finally, we reference the original Rosenblatt transform and we specify that it corresponds to a special case of our structured layers with unit gates.

We also fixed the minor typos and mistakes identified by the reviewers.

We posted detailed replies to all reviews. However, in this general reply we would like to address a major conceptual issue raised by reviewer *zyZ8*, as we think it is a central issue and its response captures the aim of our work.

---

> ### Author Response · Authors · 2021-11-16
> **General Response [2/2]**
>
> ## Issue: Why should anyone use a model instead of more complex generic layers.
>
> Question: *“Thus, a key question remains: Why not just use a more state-of-the-art flow which can handle more arbitrary distributions? Why go through the trouble of carefully specifying a probabilistic model?”*
>
> Response: Deep learning is being increasingly applied to problems in scientific fields such as physics, engineering, finance, economics, astronomy and chemistry where powerful, although often not entirely accurate, models are available. **In these contexts, users are often domain experts who find much less troublesome to use theoretical models that they already understand rather than to tinker with the many arbitrary details of deep learning architectures in order to achieve proper fit.** Consequently, the EMF approach allows to integrate the expertise of deep learning experts and domain specialists such as physicists, chemists, economists. This promotes mutually beneficial collaborations between deep learning experts and scientists and engineers. We are not the only ones suggesting this integration of deep nets and domain-knowledge, as testified by the major current trend of tailored equivariant (symmetry preserving) architectures which are being applied to many scientific fields [1,2]. Since not all models can be fully characterized in terms of their symmetries, we believe that EMFs provide a useful complementary approach.
>
> In the right settings, integrating the model in the architecture has several advantages:
>
> - If the model is approximately correct, convergence time is much faster and performance higher as training tends to converge towards local minima that preserve the structure of the model. Of course good performance can come from more complex generic architectures, but this comes with potentially very high computational cost and time spent in hyper-parameter calibration. Moreover, there is no guarantee that extra complexity will provide the appropriate inductive bias (as partially shown in our timeseries experiments with NSFs).
> - Some parts of a model are well understood theoretically but are difficult to learn from the data. For example, it is very difficult for a flow to model tail behavior, which is in contrast often easy theoretically as it involves an asymptotic regime. In this case, EMF allows to integrate the well understood asymptotic analysis with the less-understood data-driven non-asymptotic distribution.
> - The architecture becomes more interpretable for the expert user. Learned deviations from the model can be analyzed in the context of the user knowledge of the model. For example, the user can analyze the gates and see what part of the model is miss-specified. This can lead to further model development and new scientific discoveries.
>
>
>
> ## References
>
> [1] G. Kanwar, M. S. Albergo, D. Boyda, K. Cranmer, D. C. Hackett, S. Racaniere, D. J. Rezende, and P. E. Shanahan. Equivariant flow-based sampling for lattice gauge theory. Physical Review Letters, 125(12):121601, 2020.
>
> [2] V. Garcia Satorras, Emiel H., Fabian B. F., I. Posner, and M. Welling. E(n) equivariant normalizing flows for molecule generation in 3d. arXiv preprint arXiv:2105.09016, 2021.

---

> > ### Author Response · Authors · 2021-11-21
> > **General update**
> >
> > We updated the sampling time by running the samples with a GPU (tables 8, 9, 10 in appendix A5).

---

### Decision · Program_Chairs · 2022-01-20

**Decision:**

Accept (Poster)

**Comment:**

This paper proposes a method for incorporating inductive biases into the model architecture of normalizing flows through a suitable probabilistic program. All reviewers agree the paper makes an interesting contribution to the growing normalizing flow literature. The paper is well written and the idea is novel. Additionally, the experimental results are promising and the additional experiments and baselines added during the rebuttal further strengthen the paper. I recommend acceptance.